# PERSISTENT HOMOLOGY FOR HIGH-DIMENSIONAL DATA BASED ON SPECTRAL METHODS

## ABSTRACT

Persistent homology is a popular computational tool for detecting non-trivial topology of point clouds, such as the presence of loops or voids. However, many real-world datasets with low intrinsic dimensionality reside in an ambient space of much higher dimensionality. We show that in this case vanilla persistent homology becomes very sensitive to noise and fails to detect the correct topology. The same holds true for most existing refinements of persistent homology. As a remedy, we find that spectral distances on the $k$-nearest-neighbor graph of the data, such as diffusion distance and effective resistance, allow persistent homology to detect the correct topology even in the presence of high-dimensional noise. Furthermore, we derive a novel closed-form expression for effective resistance in terms of the eigendecomposition of the graph Laplacian, and describe its relation to diffusion distances. Finally, we apply these methods to several high-dimensional single-cell RNA-sequencing datasets and show that spectral distances on the $k$-nearest-neighbor graph allow robust detection of cell cycle loops.

## 1 INTRODUCTION

Algebraic topology can describe the shape of a continuous manifold. In particular, it can detect if a manifold has holes, using its so-called homology groups (Hatcher, 2002). For example, a cup has a single one-dimensional hole, or 'loop' (its handle), whereas a football has a single two-dimensional hole, or 'void' (its hollow interior). These global topological properties of an object are often helpful for understanding its overall structure. However, real-world datasets are typically given as point clouds, a discrete set of points sampled from some underlying manifold. In this setting, true homologies are trivial, as there is one connected component per point and no holes whatsoever; instead, *persistent homology* can be used to find holes in point clouds and to assign an importance score called *persistence* to each (Edelsbrunner et al., 2002; Zomorodian & Carlsson, 2004). Holes with high per-

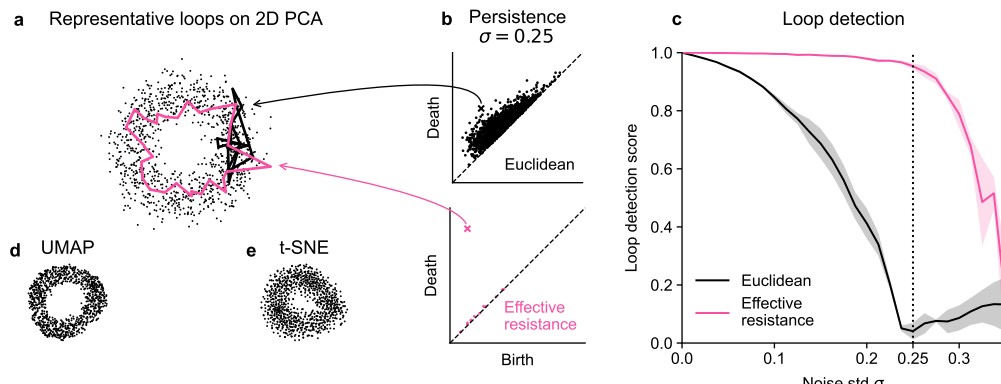

Figure 1: **a.** 2D PCA of a noisy circle ($\sigma = 0.25$, radius 1) in $\mathbb{R}^{50}$. Overlaid are representative cycles of the most persistent loops. **b.** Persistence diagrams using Euclidean distance (above) and the effective resistance (below). **c.** Loop detection scores of persistent homology using effective resistance and Euclidean distance. **d, e.** UMAP and $t$-SNE reproduced the loop structure in 2D.

sistence should be detected by persistent homology if the underlying manifold has true holes. The method has been successfully applied in various application areas such as gait recognition (Lamar-León et al., 2012), protein binding (Wang et al., 2020), nanoporous materials research (Lee et al., 2017), instance segmentation (Hu et al., 2019), and neural network analysis (Rieck et al., 2018).

Persistent homology works well for low-dimensional data (Turkes et al., 2022) but has difficulties in high dimensionality. If data points are sampled from a low-dimensional manifold embedded in a high-dimensional ambient space ('manifold hypothesis'), then the measurement noise typically affects all ambient dimensions. In this setting, vanilla persistent homology is not robust against even low levels of noise: the true topological feature can get low persistence, while noise-driven features may be more persistent. Even on a dataset as simple as a circle in $\mathbb{R}^{50}$, persistent homology based on the Euclidean distance between noisy points can fail to find the correct loop (Figure 1a – c). We are the first to systematically study the noise sensitivity of persistent homology in high dimensionality.

In contrast, dimensionality reduction methods visualizing the data in 2D, such as PCA, $t$-SNE (van der Maaten & Hinton, 2008), or UMAP (McInnes et al., 2018) are able to find and depict the loop in the same noisy dataset (Figure 1a,d,e). While such methods can be invaluable for exploring the data and generating hypotheses, they can introduce artifacts (Chari & Pachter, 2023; Wang et al., 2023) and should not be relied upon without further confirmation.

Inspired by the use of the $k$-nearest-neighbor ($k$NN) graph in modern dimensionality reduction methods (Tenenbaum et al., 2000; Roweis & Saul, 2000; Belkin & Niyogi, 2002; Hinton & Roweis, 2002; van der Maaten & Hinton, 2008; McInnes et al., 2018; Moon et al., 2019), we suggest to use persistent homology with distances based on the spectral decomposition of the $k$NN graph Laplacian, such as the effective resistance (Doyle & Snell, 1984) and the diffusion distance (Coifman & Lafon, 2006). In the same toy example as above, effective resistance succeeds in identifying the correct loop despite the high-dimensional noise (Figure 1a – c). Our contributions are:

   i. an analysis of the failure modes of persistent homology for noisy high-dimensional data;
  ii. a synthetic benchmark, with spectral distances outperforming state-of-the-art alternatives;
 iii. a closed-form expression for effective resistance, explaining its relation to diffusion distances;
 iv. an application to a range of single-cell RNA-sequencing datasets with ground-truth cycles.

## 2 RELATED WORK

Persistent homology has long been known to be sensitive to outliers (Chazal et al., 2011) and several extensions have been proposed to make it more robust. The main idea of most of these suggestions is to replace the Euclidean distance with a different distance matrix, before running persistent homology. Bendich et al. (2011) suggested to use diffusion distance (Coifman & Lafon, 2006), but their empirical validation was limited to a single dataset in 2D. Anai et al. (2020) suggested to use distance-to-measure (DTM) (Chazal et al., 2011) and Fernández et al. (2023) proposed to use Fermat distances (Groisman et al., 2022). Vishwanath et al. (2020) introduced persistent homology based on robust kernel density estimation, an approach that itself becomes challenging in high dimensionality. All of these works focused mostly on low-dimensional datasets ($<$10D, mostly 2D or 3D), while our work specifically addresses the challenges of persistent homology in high dimensions.

Below, we will recommend using effective resistance and diffusion distances for persistent homology in high-dimensional spaces. Both of these distances, as well as the shortest path distance, have been used in combination with persistent homology to analyze the topology of graph data (Petri et al., 2013; Hajij et al., 2018; Aktas et al., 2019; Mémoli et al., 2022). Shortest paths on the $k$NN graph were also used by Naitzat et al. (2020) and Fernández et al. (2023). Motivated by the performance of UMAP (McInnes et al., 2018) for dimensionality reduction, Gardner et al. (2022) and Hermansen et al. (2022) used UMAP affinities to define distances for persistent homology.

Effective resistance is a well-established graph distance (Doyle & Snell, 1984; Fouss et al., 2016). A correction to effective resistance, more appropriate for large graphs, was suggested by von Luxburg et al. (2010a) and von Luxburg et al. (2014). When speaking of *effective resistance*, we mean this corrected version, if not otherwise stated. It has not been combined with persistent homology. Conceptually similar diffusion distances (Coifman & Lafon, 2006) have been used in single-cell RNA-sequencing data analysis, for dimensionality reduction (Moon et al., 2019), trajectory infer-

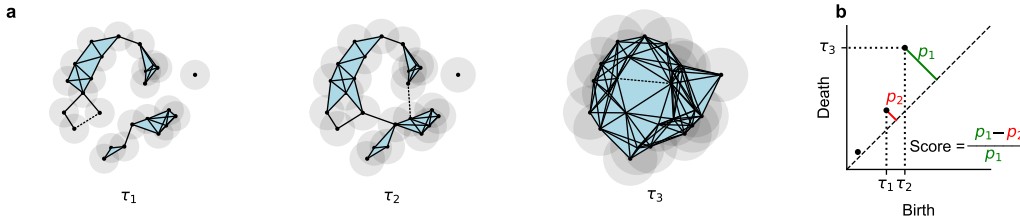

Figure 2: **a.** Persistent homology applied to a noisy circle ($n = 25$) in 2D. Dotted lines show the edges that lead to the birth / death of two loops (Section 3). **b.** The corresponding persistence diagram with three detected holes. Our *hole detection score* measures the gap in persistence between the first and the second detected holes (Section 7.1.1).

ence (Haghverdi et al., 2015), feature extraction (Chew et al., 2022), and hierarchical clustering, similar to 0D persistent homology (Brugnone et al., 2019; Kuchroo et al., 2023).

Persistent homology has been applied to single-cell RNA-sequencing data, but only the concurrent work of Flores-Bautista & Thomson (2023) applies it directly to the high-dimensional data. Wang et al. (2023) used a Witness complex on a PCA of the data. Other works applied persistent homology to a derived graph, e.g., a gene regulator network (Masoomy et al., 2021) or a Mapper graph (Singh et al., 2007; Rizvi et al., 2017). In other biological contexts, persistent homology has also been applied to a low-dimensional representation of the data: 3D PCA of cytometry data (Mukherjee et al., 2022), 6D PCA of hippocampal spiking data (Gardner et al., 2022), and 3D PHATE embedding of calcium signaling (Moore et al., 2023). Several recent applications of persistent homology only computed 0D features (i.e. clusters) (Hajij et al., 2018; Jia & Chen, 2022; Petenkaya et al., 2022), which amounts to doing single linkage clustering (Gower & Ross, 1969). Here we only investigate the detection of higher-dimensional (1D and 2D) holes with persistent homology.

## 3 BACKGROUND: PERSISTENT HOMOLOGY

Persistent homology computes the homology of a space at different scales. For point clouds, the different scales are often given by growing a ball around each point (Figure 2a), and letting the radius $\tau$ grow from 0 to infinity. For each value of $\tau$, homology groups of the union of all balls are computed to find the holes, and holes that *persist* for longer time periods $\Delta\tau$ are considered more prominent. Note that at $\tau \approx 0$, there are no holes as the balls are non-overlapping, while at sufficiently large $\tau \gg 0$ there are no holes as all the balls merge together.

To keep the computation tractable, instead of the union of growing balls, persistent homology operates on a so-called *filtered simplicial complex* (Figure 2a). A simplicial complex is a hypergraph containing points as nodes, edges between nodes, triangles bounded by edges, and so forth. These building blocks are called *simplices*. At time $\tau$, the hypergraph encodes all intersections between the balls and suffices to find the holes. The complexes at smaller $\tau$ values are nested within the complexes at larger $\tau$ values, and together form a filtered simplicial complex, with $\tau$ being the filtration parameter (filtration time). In this work, we only use the Vietoris–Rips complex, which includes an $n$-simplex $(v_0, v_1, \ldots, v_n)$ at filtration time $\tau$ if the distance between all pairs $v_i, v_j$ is at most $\tau$. Therefore, to build a Vietoris–Rips complex, it suffices to provide pairwise distances between all pairs of points. We compute persistent homology via the `ripser` pckage (Bauer, 2021).

Persistent homology consists of a set of holes for each dimension. We limit ourselves to loops and voids. Each hole has associated birth and death times $(\tau_b, \tau_d)$, i.e., the first and last filtration value $\tau$ at which that hole exists. Their difference $p = \tau_d - \tau_b$ is called the *persistence* or *life time* of the hole and quantifies its prominence. The birth and death times can be visualized as a scatter plot (Figure 2b), known as the *persistence diagram*. Points far from the diagonal have high persistence.

This process is illustrated in Figure 2 for a noisy sample of $n = 25$ points from a $S^1 \subset \mathbb{R}^2$. At $\tau_1$ a small spurious loop is formed thanks to the inclusion of the dotted edge, but it dies soon afterwards. The ground-truth loop is formed at $\tau_2$ and dies at $\tau_3$, once the hole is completely filled in by triangles. All three loops (one-dimensional holes) found in this dataset are shown in the persistence diagram.

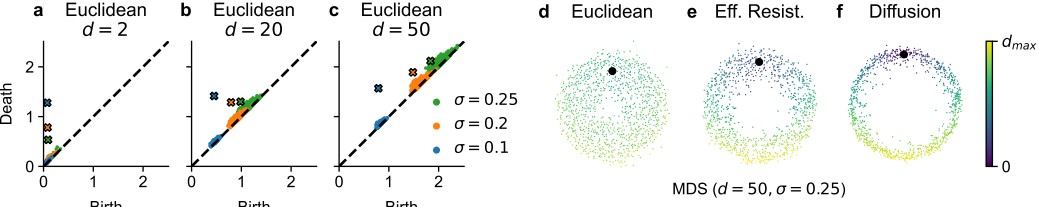

Figure 3: **a – c.** Persistence diagrams of a noisy circle in different ambient dimensionality and with different amount of noise. **d – f.** Multidimensional scaling of Euclidean, effective resistance and diffusion distance for a noisy circle in $\mathbb{R}^{50}$. Color indicates the distance to the highlighted point.

## 4    THE CURSE OF DIMENSIONALITY FOR PERSISTENT HOMOLOGY

While persistent homology has been shown to be robust to small changes in the point positions (Cohen-Steiner et al., 2005), the curse of dimensionality can still severely hurt its performance.

To illustrate, we consider the same toy setting as in Figure 1: we sample points from $S^1 \subset \mathbb{R}^d$, and add Gaussian noise of standard deviation $\sigma$ to each of the ambient coordinates. When $d = 2$, higher noise does not affect the birth times but leads to lower death times (Figure 3a), because some points get distorted to the middle of the circle and let the hole fill up at earlier $\tau$. When we increase the ambient dimensionality to $d = 20$, higher noise leads to later birth times (Figure 3b) because the distances are now dominated by the noise dimensions rather than by the circular structure.[1] Finally, for $d = 50$ both the birth *and* the death times increase with $\sigma$ (Figure 3c), such that the ground-truth hole disappears in the cloud of spurious holes.

In other words, in a high-dimensional space, all pairwise distances become similar, and the circular structure fails to stand out. Applying MDS to the Euclidean distance matrix obtained with $d = 50$ and $\sigma = 0.25$ yields a 2D embedding with almost no visible hole (Figure 3d). This is a 2D shadow of the fact that the distances due to noise dominate the distances due to the circular structure. Therefore, the failure modes of persistent homology differ between low- and high-dimensional spaces: While in low dimensions, persistent homology is susceptible to outlier points in the middle of the circle, in high dimensions, there are no points in the middle of the circle; instead, all distances become too similar, hiding the true loops. See Appendix S6 for more details on the effect of outliers.

## 5    BACKGROUND: EFFECTIVE RESISTANCE AND DIFFUSION DISTANCES

Many modern manifold learning and dimensionality reduction methods rely on the $k$-nearest-neighbor ($k$NN) graph of the data. This works well because although distances become increasingly similar in high-dimensional spaces, nearest neighbors still carry information about the data manifold. To make persistent homology overcome high-dimensional noise, we therefore suggest to rely on the $k$NN graph. A natural choice is to use its geodesics, but as we show below this does not work well, likely because a single graph edge across a circle can make the corresponding feature die too early. Instead, we propose to use spectral methods, such as the effective resistance or diffusion distance. Both methods rely on random walks and thus integrate information about all edges.

For a connected graph $G$ with $n$ nodes, e.g., a symmetric $k$NN graph, let $A$ be its symmetric, $n \times n$ adjacency matrix with elements $a_{ij} = 1$ if edge $ij$ exits in $G$ and $a_{ij} = 0$ otherwise. Then the degree matrix $D$ is defined by $D = \text{diag}\{d_i\}$, where $d_i = \sum_{j=1}^{n} a_{ij}$ are the node degrees. We define $\text{vol}(G) = \sum_{i=1}^{n} d_i$. Let $H_{ij}$ be the *hitting time* from node $i$ to $j$, i.e., the average number of edges it takes a random walker, that starts at node $i$ randomly moving along edges, to reach node $j$. Then the naive effective resistance is defined as $\tilde{d}_{ij}^{\text{eff}} = (H_{ij} + H_{ji})/\text{vol}(G)$. This naive version is known to be unsuitable for large graphs (Figure S7) because it reduces to $\tilde{d}_{ij}^{\text{eff}} \approx 1/d_i + 1/d_j$ (von

---

[1]It is easy to see that the expected squared distance between two random samples from a $d$-dimensional isotropic Gaussian with standard deviation $\sigma$ is $2d\sigma^2$. The non-squared distance grows with $\sqrt{d}$ (Appendix B).

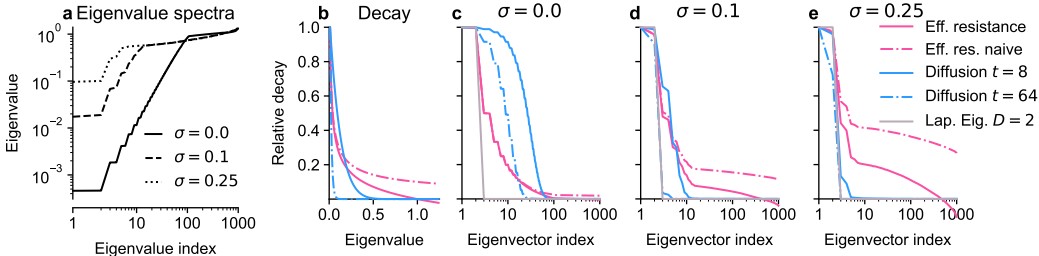

Figure 4: **a.** Eigenvalue spectra of the s$k$NN graph Laplacian for the noisy circle in ambient $\mathbb{R}^{50}$ for noise levels $\sigma = \{0.0, 0.1, 0.25\}$. **b.** Decay of eigenvector contribution based on the eigenvalue for effective resistance and diffusion distances. **c – e.** Relative contribution of each eigenvector for effective resistance, diffusion distance and Laplacian Eigenmaps for various noise levels (Section 6).

Luxburg et al., 2010a). Therefore we used von Luxburg et al. (2010a)'s corrected version

$$d_{ij}^{\mathrm{eff}} = \tilde{d}_{ij}^{\mathrm{eff}} - 1/d_i - 1/d_j + 2a_{ij}/(d_i d_j) - a_{ii}/d_i^2 - a_{jj}/d_j^2. \tag{1}$$

Diffusion distances also rely on random walks. The random walk transition matrix is given by $P = D^{-1}A$. Then $P_{i,:}^t$, the $i$-th row of $P^t$, holds the probability distribution over nodes after $t$ steps of a random walker starting at node $i$. The diffusion distance is then defined as

$$d_{ij}(t) = \sqrt{\mathrm{vol}(G)}\|(P_{i,:}^t - P_{j,:}^t)D^{-1}\|. \tag{2}$$

There are many possible random walks between nodes $i$ and $j$ if they both reside in the same densely connected region of the graph, while it is unlikely for a random walker to cross between sparsely connected regions. As a result, both effective resistance and diffusion distance are small between parts of the graph that are densely connected and are robust against single stray edges. Indeed, the MDS embedding of the effective resistance and of the diffusion distance of the circle in ambient $\mathbb{R}^{50}$ both clearly show the circular structure (Figure 3e,f).

## 6 RELATION BETWEEN SPECTRAL DISTANCES

Laplacian Eigenmaps distance and diffusion distance can be written as Euclidean distances in data representations given by appropriately scaled eigenvectors of the graph Laplacian. In this section we derive a similar closed-form formula for effective resistance.

Using the definitions from Section 5, let $v_1, \ldots, v_n$ denote the eigenvectors of the Laplacian $L = D - A$ and $\lambda_1, \ldots, \lambda_n$ their eigenvalues. Let further $A^{\mathrm{sym}} = D^{-\frac{1}{2}}AD^{-\frac{1}{2}}$ and $L^{\mathrm{sym}} = I - A^{\mathrm{sym}}$ be the symmetrically normalized adjacency matrix and the symmetrically normalized graph Laplacian. We denote the eigenvectors of $L^{\mathrm{sym}}$ by $u_1, \ldots, u_n$ and their eigenvalues by $\mu_1, \ldots, \mu_n$ in increasing order. For a connected graph, $\lambda_1 = \mu_1 = 0$ and $v_1 = u_1 = (1, \ldots, 1)/\sqrt{n}$.

The $D$-dimensional Laplacian Eigenmaps embedding is given by the first $D$ nontrivial eigenvectors:

$$d_{ij}^{\mathrm{LE}}(D) = \|e_i^{\mathrm{LE}}(D) - e_j^{\mathrm{LE}}(D)\|, \text{ where } e_i^{\mathrm{LE}}(D) = (v_{2,i}, \ldots, v_{(D+1),i}). \tag{3}$$

The diffusion distance for $t$ diffusion steps is given by (Coifman & Lafon, 2006)

$$d_{ij}^{\mathrm{diff}}(t) = \|e_i^{\mathrm{diff}}(t) - e_j^{\mathrm{diff}}(t)\|, \text{ where } e_i^{\mathrm{diff}}(t) = \frac{1}{\sqrt{d_i}} \left((1 - \mu_2)^t u_{2,i}, \ldots, (1 - \mu_n)^t u_{n,i}\right). \tag{4}$$

Finally, the original uncorrected version of effective resistance is given by (Fouss et al., 2007)

$$\tilde{d}_{ij}^{\mathrm{eff}} = \|\tilde{e}_i^{\mathrm{eff}} - \tilde{e}_j^{\mathrm{eff}}\|^2, \text{ where } \tilde{e}_i^{\mathrm{eff}} = \left(\frac{1}{\sqrt{\lambda_2}}v_{2,i}, \ldots, \frac{1}{\sqrt{\lambda_n}}v_{n,i}\right). \tag{5}$$

Here we show that the corrected effective resistance (von Luxburg et al., 2010a) can also be written in this form (see the proof in Appendix A):

**Proposition 1.** *The corrected effective resistance distance can be computed by*

$$d_{ij}^{\mathrm{eff}} = \|e_i^{\mathrm{eff}} - e_j^{\mathrm{eff}}\|^2, \text{ where } e_i^{\mathrm{eff}} = \frac{1}{\sqrt{d_i}} \left(\frac{1 - \mu_2}{\sqrt{\mu_2}}u_{2,i}, \ldots, \frac{1 - \mu_n}{\sqrt{\mu_n}}u_{n,i}\right). \tag{6}$$

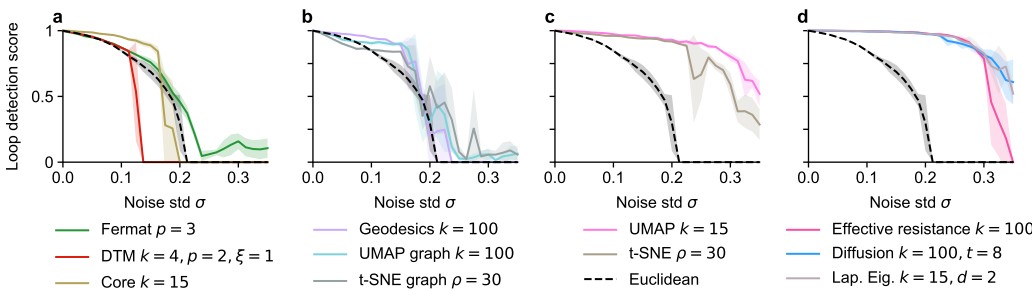

Figure 5: Loop detection score for persistent homology with various distances on a noisy circle in $\mathbb{R}^{50}$. The best hyperparameter setting for each distance is shown. Methods are grouped for clarity.

As we operate on the symmetric $k$NN graph (s$k$NN), which contains edge $ij$ if $i$ is among the $k$ nearest neighbors of $j$ or vice versa, the degree distribution is close to constant. As a result, the $1/\sqrt{d_i}$ terms have little effect and the eigenvectors and the relative size of the eigenvalues do not depend much on the normalization of the Laplacian. Note that effective resistance is a *squared* Euclidean distance. However, omitting the square amounts to taking the square root of all birth and death times, maintaining the loop detection performance of effective resistance (Figure S7). Therefore, the main difference between the spectral methods boils down to how they decay higher eigenvectors based on the corresponding eigenvalues.

The naive effective resistance decays the eigenvectors with $1/\sqrt{\lambda_i}$, which is much slower than diffusion distances' $(1 - \mu_i)^t$ for ($t \in [8, 64]$), while corrected effective resistance shows intermediate behaviour (Figure 4b). The correction matters little for the $S^1 \subset \mathbb{R}^{50}$ in the absence of noise, when the first eigenvalues are much smaller than the rest and dominate the embedding (Figure 4a,c) but becomes important as the noise and consequently the low eigenvalues increase (Figure 4a,d,e). As the noise increases, the decay for diffusion distances gets closer to a step function preserving only the first two non-constant eigenvectors, similar to Laplacian Eigenmaps with $D = 2$ (Figure 4c – e).

## 7 SPECTRAL DISTANCES FIND HOLES IN HIGH-DIMENSIONAL SPACES

We benchmark various distances as input to persistent homology in high ambient dimensionality.

**Distance measures** We examined twelve other distances as input to persistent homology, beyond the Euclidean distance. Full definitions are given in Appendix C. First, there are some state-of-the-art approaches for persistent homology in presence of noise and outliers. Fermat distances (Fernández et al., 2023) aim to exaggerate large over small distances to incorporate the density of the data. Distance-to-measure (DTM) (Anai et al., 2020) aims for outlier robustness by combining the Euclidean distance with the distances from each point to its $k$ nearest neighbors, which are high for low-density outliers. Similarly, the core distance used in the HDBSCAN algorithm (Campello et al., 2015; Damm, 2022) raises each Euclidean distance at least to the distance between incident points and their $k$-th nearest neighbors. We evaluate these methods here with respect to Gaussian noise in high-dimensional ambient space, a different noise model than the one for which these method were designed. Second, we consider some non-spectral graph distances. The geodesic distance on the $k$NN graph was popularized by Isomap (Tenenbaum et al., 2000) and used for persistent homology by Naitzat et al. (2020). Following Gardner et al. (2022) we used distances based on UMAP affinities, and also experimented with $t$-SNE affinities. Third, we computed $t$-SNE and UMAP embeddings and used distances in the 2D embedding space. Finally, we explored methods using the spectral decomposition of the $k$NN graph Laplacian, see Section 6: (corrected) effective resistance, diffusion distance, and the distance in Laplacian Eigenmaps' embedding space.

All methods come with hyperparameters. We report the results for the best hyperparameter setting on each dataset. The selected hyperparameter values can be found in Appendix E.

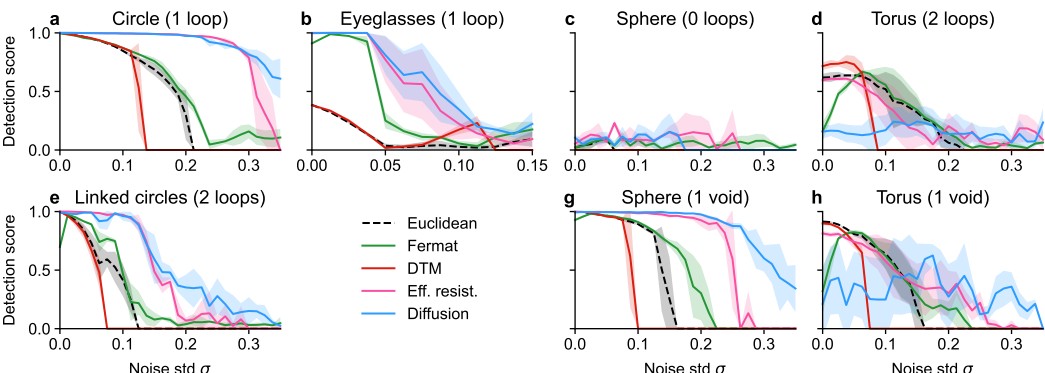

Figure 6: Loop detection score for selected methods on all synthetic datasets in ambient $\mathbb{R}^{50}$. More methods and hyperparameter settings can be found in Figures S10 – S19.

**Performance metric** The output of persistent homology is a persistence diagram showing birth and death times for all detected holes. It may be difficult to decide whether this procedure has actually 'detected' a hole in the data. Ideally, for a dataset with $m$ ground-truth holes, the persistence diagram should have $m$ points with high persistence while all other points should have low persistence and lie close to the diagonal. Therefore, for $m$ ground-truth features, our metric $s_m \in [0, 1]$, which we call the *hole detection score*, is the relative gap between the persistences $p_m$ and $p_{m+1}$ of the $m$-th and $(m + 1)$-th most persistent features: $s_m = (p_m - p_{m+1})/p_m$. This corresponds to the visual gap between them in the persistence diagram (Figure 2b).

In addition, we set $s_m = 0$ if all features in the persistence diagram have very low death-to-birth ratios $\tau_d/\tau_b < 1.25$. This handles situations with very few detected holes that die very quickly after being born, which otherwise can have spuriously high $s_m$ values. This was done everywhere apart from the qualitative Figures 1, 8 and in Figure S12. We call this heuristic thresholding.

We report the mean over three random seeds. Shading and error bars indicate the standard deviation.

## 7.1 SYNTHETIC BENCHMARK

### 7.1.1 BENCHMARK SETUP

In our synthetic benchmark, we evaluated the performance of various distance measures in conjunction with persistent homology on five manifolds: a circle, a pair of linked circles, the eyeglasses dataset (a circle squeezed nearly to a figure eight) (Fernández et al., 2023), the sphere, and the torus. The radii of the circles, the sphere, and the torus' tube were set to 1, the bottleneck of the eyeglasses was 0.7, and the torus' tube followed a circle of radius 2. In each case, we uniformly sampled $n = 1\,000$ points from the manifold, mapped them isometrically to $\mathbb{R}^d$ for $d \in [2, 50]$, and then added isotropic Gaussian noise sampled from $\mathcal{N}(\mathbf{0}, \sigma^2 \mathbf{I}_d)$ for $\sigma \in [0, 0.35]$. More details can be found in Appendix D. For each resulting dataset, we computed persistent homology for loops and, for the sphere and the torus, also for voids. We never computed holes of dimension 3 or higher.

### 7.1.2 RESULTS ON SYNTHETIC DATA

On the circle dataset in $\mathbb{R}^{50}$, persistent homology with all distance metrics found the correct hole when the noise level $\sigma$ was very low (Figure 5). However, as the amount of noise increased, the performance of Euclidean distance quickly deteriorated, reaching near-zero score at $\sigma \approx 0.25$. Most other distances outperformed the Euclidean distance, at least in the low noise regime. Fermat distance did not have any effect, and neither did DTM distance, which collapsed at $\sigma \approx 0.15$ due to thresholding (Figure 5a). Geodesics, UMAP/$t$-SNE graph, and core distance offered only a modest improvement over Euclidean (Figure 5b). In contrast, embedding-based distances performed very well on the circle (Figure 5c), but have obvious *a priori* limitations: for example, a 2D embedding cannot possibly have a void. Finally, all spectral methods (effective resistance, diffusion, and Laplacian Eigenmaps) showed similarly excellent performance (Figure 5d).

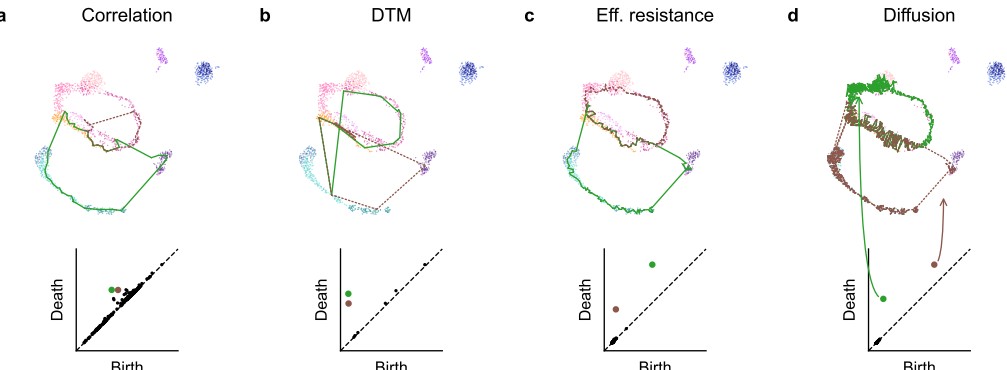

Figure 7: **a.** Loop detection score of various methods on a noisy circle depending on the ambient dimensionality. Noise $\sigma = 0.25$. **b – e.** Heat maps for $\sigma \in [0, 0.35]$ and $d \in [2, 50]$.

In line with these results, spectral methods outperformed other methods across most synthetic datasets in $\mathbb{R}^{50}$ (Figure 6). DTM collapsed earlier than Euclidean but detected loops on the torus for low noise levels best by a small margin. Fermat distance typically had little effect and provided a benefit over Euclidean only on the eyeglasses and the sphere. Spectral distances outperformed all other methods on all datasets apart from the torus, where effective resistance was on par with Euclidean but diffusion performed poorly. On a more densely sampled torus, all methods performed better and the spectral methods again outperformed the others (Figure S18). On all other datasets diffusion had a small edge over effective resistance for large $\sigma$. Reassuringly, all methods passed the negative control and did not find any persistent loops on the sphere (Figure 6e).

As discussed in Section 4, persistent homology with Euclidean distances deteriorates with increasing ambient dimensionality. Using the circle data in $\mathbb{R}^{d}$, we found that if the noise level was fixed at $\sigma = 0.25$, no persistent loop was found using Euclidean distances for $d \gtrsim 30$ (Figure 7). In the same setting, DTM deteriorated even more quickly than Euclidean distances. In contrast, effective resistance and diffusion distance were robust against both the high noise level and the large ambient dimension (Figure 7a, c – e). For more details, compare Figures S10 and S11.

### 7.2 DETECTING CYCLES IN SINGLE-CELL RNA-SEQUENCING DATA

We applied our methods to six single-cell RNA-sequencing datasets: Malaria (Howick et al., 2019), Neurosphere, and Hippocampus from (Zheng et al., 2022), HeLa2 (Schwabe et al., 2020), Neural IPCs (Braun et al., 2022), and Pancreas (Bastidas-Ponce et al., 2019). Single-cell RNA-sequencing data consists of expression levels for thousands of genes in individual cells, so the data is high-dimensional and notoriously noisy. All selected datasets are known to contain circular structures, usually corresponding to the cell division cycle. In each case, we followed existing preprocessing pipelines leading to representations with $10$ to $5\,156$ dimensions. Datasets with more than $4\,000$ cells were downsampled to $n = 1\,000$ (Appendix D).

Figure 8: Malaria dataset. **a – d.** Representatives of the two most persistent loops overlaid on UMAP embedding (top) and persistence diagrams (bottom) using four methods. Biology dictates that there should be two loops (in warm colors and in cold colors) connected as in a figure eight.

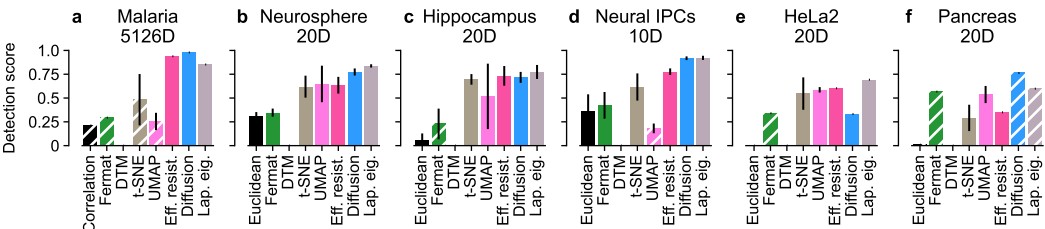

Figure 9: Loop detection scores on six high-dimensional scRNA-seq datasets. Hatched bars indicate implausible representatives. See Figure S20 for detection scores for different hyperparameter values.

The Malaria dataset is expected to contain two cycles: the parasite replication cycle in red blood cells, and the parasite transmission cycle between human and mosquito hosts. Following Howick et al. (2019), we based all computations for this dataset (and all derived distances) on the correlation distance instead of the Euclidean distance. Persistent homology based on the correlation distance itself failed to correctly identify the two ground-truth cycles and DTM produced representatives that only rough approximate the two ground truth cycles (Figure 8a,b). But both effective resistance and diffusion distance successfully uncovered both cycles, with $s_2 > 0.9$ (Figure 8 c,d).

Across all six datasets, the detection scores were higher for spectral methods than for their competitors (Figure 9). Furthermore, we manually investigated representative loops for all considered methods on all datasets and found several cases where the most persistent loop(s) was/were likely not correct (hatched bars in Figure 9). Overall, we found that the spectral methods, and in particular effective resistance, could reliably find the correct loops with high detection score. Persistent homology based on the $t$-SNE and UMAP embeddings could also often identify the correct loop structure and on average worked better than vanilla persistent homology, Fermat distances, and DTM.

## 8 DISCUSSION

In this work we asked how to use persistent homology on high-dimensional noisy datasets. We demonstrated that, as the dimensionality of the data increases, the main problem for persistent homology shifts from handling outliers to handling noise dimensions (Section 4, Appendix G). We used a synthetic benchmark to show that vanilla persistent homology and many of its existing extensions struggle to find the correct topology in this setting, whereas spectral methods based on the $k$NN graph, such as the effective resistance and diffusion distances, work well (Section 7.1). We view it as an advantage that these existing methods can handle the important problem of high-dimensional noise. We derived an expression for effective resistance based on the eigendecomposition of the graph Laplacian, relating it to diffusion distances and Laplacian Eigenmaps (Section 6). Finally, we showed that spectral distances outperform all competitors on single-cell data (Section 7.2).

In the real-world applications, it was important to look at representatives of detected holes as sometimes methods found persistent, but arguably incorrect loops. That said, each hole homology class has many different representative cycles, making interpretation difficult. Given ground-truth cycles, an automatic procedure for evaluating cycle correctness remains an interesting research question.

Dimensionality reduction methods are designed to handle high-dimensional data. In the case of $t$-SNE and UMAP, we observed that persistent homology based on the 2D embeddings performed much better than using their s$k$NN graph affinities, underlining that the key to the success of these methods is in their embedding optimization rather than their notion of similarity (Böhm et al., 2022; Damrich & Hamprecht, 2021). In contrast, spectral distances on the s$k$NN graph worked well without a low-dimensional embedding (Sections 7.1, 7.2).

One limitation of persistent homology is its computational complexity. It scales as $\mathcal{O}(n^{3(\delta+1)})$ for $n$ points and topological holes of dimension $\delta$ (Myers et al., 2023). This aggravates other problems of high-dimensional datasets as dense sampling in high-dimensional space would require a prohibitively large sample size. When combining persistent homology with non-Euclidean distance measures, the approach of Bendich et al. (2011), who performed subsampling after computation of the distance matrix, is particularly attractive, and forms an interesting avenue for future research.

## REPRODUCIBILITY STATEMENT

We prove our theoretical statements in Appendices A and B. We describe the methods and datasets used in detail in Appendices C and D. The hyperparameters for the reported results can be found in Appendix E. Finally, we give details on our implementation and hardware in Appendix F. Our complete code is part of the supplementatry material.

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

SUPPLEMENTARY TEXT

## A  EFFECTIVE RESISTANCE AS SPECTRAL METHOD

Let $G$ be a weighted, connected graph of $n$ nodes. Denote by $A = (a_{ij})_{i,j=1,\dots,n}$ the weighted adjacency matrix whose entries $a_{ij} = a_{ji}$ equal the edge weight of edge $ij$ if edge $ij$ is part of the graph and zero otherwise. Further denote by $D = \text{diag}(d_i)$ the degree matrix, where $d_i = \sum_{j=1,\dots,n} a_{ij}$ are the degrees. We define $\text{vol}(G) = \sum_i d_i$. Let further $A^{\text{sym}} = D^{-\frac{1}{2}} A D^{-\frac{1}{2}}$ and $L^{\text{sym}} = I - A^{\text{sym}}$ be the symmetrically normalized adjacency matrix and the symmetrically normalized graph Laplacian. Denote the eigenvectors of $L^{\text{sym}}$ by $u_1, \dots, u_n$ and their eigenvalues by $\mu_1, \dots, \mu_n$ in increasing order. The eigenvectors and eigenvalues of $A^{\text{sym}}$ are $u_1, \dots, u_n$ and $1 - \mu_1, \dots, 1 - \mu_n$.

**Definition 2.** The effective resistance distance between nodes $i$ and $j$ is defined as

$$\tilde{d}_{ij}^{\text{eff}} = \frac{1}{\text{vol}(G)}(H_{ij} + H_{ji}), \tag{7}$$

where $H_{ij}$ is the hitting time from $i$ to $j$, i.e., the expected number of steps that a random walker starting at node $i$ takes to reach node $j$ for the first time.

The corrected effective resistance distance between nodes $i$ and $j$ is defined as

$$d_{ij}^{\text{eff}} = \tilde{d}_{ij}^{\text{eff}} - \frac{1}{d_i} - \frac{1}{d_j} + 2\frac{a_{ij}}{d_i d_j} - \frac{a_{ii}}{d_i^2} - \frac{a_{jj}}{d_j^2}. \tag{8}$$

The following proposition is an elaboration of Prop. 4 in von Luxburg et al. (2010a).

**Proposition 3.** *The corrected effective resistance distance $d_{ij}^{\text{eff}}$ between nodes $i$ and $j$ can be computed by $d_{ij}^{\text{eff}} = \|e_i - e_j\|^2$, where*

$$e_i = \frac{1}{\sqrt{d_i}}\left(\frac{1-\mu_2}{\sqrt{\mu_2}}u_{2,i}, \dots, \frac{1-\mu_n}{\sqrt{\mu_n}}u_{n,i}\right).$$

*Proof.* By the fourth step of the large equation in the proof of Proposition 2 in von Luxburg et al. (2010b), we have

$$\frac{1}{\text{vol}(G)}H_{ij} = \frac{1}{d_j} + \langle b_j, A^{\text{sym}}(b_j - b_i)\rangle + \sum_{r=2}^{n}\frac{1}{\mu_r}\langle A^{\text{sym}}b_j, u_r(u_r)^T(A^{\text{sym}}(b_j - b_i))\rangle, \tag{9}$$

where $b_i = \frac{1}{\sqrt{d_i}}e_i = D^{-\frac{1}{2}}e_i$ with $e_i$ the $i$-th standard basis vector.

Adding this expression for $ij$ and $ji$ and using the definition of $\tilde{d}_{ij}^{\text{eff}}$, we get

$$\tilde{d}_{ij}^{\text{eff}} - \frac{1}{d_i} - \frac{1}{d_j} = \frac{1}{\text{vol}(G)}(H_{ij} + H_{ji}) - \frac{1}{d_i} - \frac{1}{d_j} \tag{10}$$

$$= \langle b_j, A^{\text{sym}}(b_j - b_i)\rangle + \langle b_i, A^{\text{sym}}(b_i - b_j)\rangle \tag{11}$$

$$+ \sum_{r=2}^{n}\frac{1}{\mu_r}\langle A^{\text{sym}}b_j, u_r(u_r)^T(A^{\text{sym}}(b_j - b_i))\rangle \tag{12}$$

$$+ \sum_{r=2}^{n}\frac{1}{\mu_r}\langle A^{\text{sym}}b_i, u_r(u_r)^T(A^{\text{sym}}(b_i - b_j))\rangle \tag{13}$$

$$= \langle b_j - b_i, A^{\text{sym}}(b_j - b_i)\rangle \tag{14}$$

$$+ \sum_{r=2}^{n}\frac{1}{\mu_r}\langle A^{\text{sym}}(b_j - b_i), u_r(u_r)^T(A^{\text{sym}}(b_j - b_i))\rangle \tag{15}$$

For ease of exposition, we treat both summands separately. By unpacking the definitions and symmerty of $D$, we get

$$\langle b_j - b_i, A^{\text{sym}}(b_j - b_i)\rangle = \langle D^{-\frac{1}{2}}(e_j - e_i), A^{\text{sym}}D^{-\frac{1}{2}}(e_j - e_i)\rangle \tag{16}$$

$$= \langle(e_j - e_i), D^{-1}AD^{-1}(e_j - e_i)\rangle \tag{17}$$

$$= \frac{a_{jj}}{d_j^2} - 2\frac{a_{ij}}{d_i d_j} + \frac{a_{ii}}{d_i^2} \tag{18}$$

Since the $u_r$ are eigenvectors of $A^{\text{sym}}$ with eigenvalue $1 - \mu_r$ and $A^{\text{sym}}$ is symmetric, we also get

$$\sum_{r=2}^{n} \frac{1}{\mu_r} \langle A^{\text{sym}}(b_j - b_i), u_r(u_r)^T(A^{\text{sym}}(b_j - b_i)) \rangle \tag{19}$$

$$= \sum_{r=2}^{n} \frac{1}{\mu_r} \langle (b_j - b_i), (A^{\text{sym}}u_r)(A^{\text{sym}}u_r)^T(b_j - b_i) \rangle \tag{20}$$

$$= \sum_{r=2}^{n} \frac{1}{\mu_r} \langle (b_j - b_i), (1 - \mu_r)u_r((1 - \mu_r)u_r)^T(b_j - b_i) \rangle \tag{21}$$

$$= \sum_{r=2}^{n} \left( \frac{1 - \mu_r}{\sqrt{\mu_r}} u_r^T D^{-\frac{1}{2}}(e_j - e_i) \right)^2 \tag{22}$$

$$= \left\| \frac{1}{\sqrt{d_j}} \left( \frac{1 - \mu_r}{\sqrt{\mu_r}} u_{r,j} \right)_{r=2,\ldots,n} - \frac{1}{\sqrt{d_i}} \left( \frac{1 - \mu_r}{\sqrt{\mu_r}} u_{r,i} \right)_{r=2,\ldots,n} \right\|^2 \tag{23}$$

$$= \| e_j - e_i \|^2 \tag{24}$$

Putting everything together yields the result

$$d_{ij}^{\text{eff}} = \tilde{d}_{ij}^{\text{eff}} - \frac{1}{d_i} - \frac{1}{d_j} + 2\frac{a_{ij}}{d_i d_j} - \frac{a_{ii}}{d_i^2} - \frac{a_{jj}}{d_j^2} \tag{25}$$

$$= \frac{a_{jj}}{d_j^2} - 2\frac{a_{ij}}{d_i d_j} + \frac{a_{ii}}{d_i^2} + \|e_j - e_i\|^2 + 2\frac{a_{ij}}{d_i d_j} - \frac{a_{ii}}{d_i^2} - \frac{a_{jj}}{d_j^2} \tag{26}$$

$$= \| e_j - e_i \|^2. \tag{27}$$

$\square$

## B  PAIRWISE NOISE DISTANCES IN HIGH-DIMENSIONAL SPACES

Consider two independent random variables $X$ and $Y$, both following a $d$-dimensional spherical normal distribution: $X, Y \sim \mathcal{N}(\mathbf{0}, \sigma^2 \mathbf{I}_d)$. Their difference $X - Y$ is also normally distributed $X - Y \sim \mathcal{N}(\mathbf{0}, 2\sigma^2 \mathbf{I}_d)$, so the expected squared distance is simply $\mathbb{E}(\|X - Y\|^2) = 2\sigma^2 d$.

However, persistent homology operates on non-squared distances, which is why we are interested in $\mathbb{E}(\|X - Y\|)$. Using a somewhat more involved calculation, it can be shown that it scales as $\sqrt{2}\sigma d$.

**Proposition 4.** *Let $X$ and $Y$ be isotropic $d$-dimensional normally distributed random variables, $X, Y \sim \mathcal{N}(\mathbf{0}, \sigma^2 \mathbf{I}_d)$, where $\mathbf{I}_d$ is the $d$-dimensional identity matrix and $\sigma > 0$. Then the expected distance between $X$ and $Y$ is*

$$\mathbb{E}(\|X - Y\|) = 2\sigma \frac{\Gamma(\frac{d+1}{2})}{\Gamma(\frac{d}{2})} = 2\sigma \left( \sqrt{\frac{d}{2}} + O(d^{-0.5}) \right). \tag{28}$$

*Proof.* The random variable $X - Y$ is also normally distributed, $X - Y \sim \mathcal{N}(\mathbf{0}, 2\sigma^2 \mathbf{I}_d)$. Therefore, $\frac{\|X-Y\|}{\sqrt{2}\sigma}$ is Chi-distributed, with mean $\sqrt{2}\frac{\Gamma(\frac{d+1}{2})}{\Gamma(\frac{d}{2})}$. This shows the first part of the claim.

For the approximate expression, we only treat the case of even $d$ for ease of exposition. Let $d = 2d'$. Then by properties of the Gamma function and the binomial coefficient, we have

$$\frac{\Gamma(\frac{d+1}{2})}{\Gamma(\frac{d}{2})} = \frac{\Gamma(d' + 0.5)}{\Gamma(d')} = \frac{\sqrt{\pi}\frac{(2d'-1)!!}{2^{d'}}}{(d'-1)!} = \sqrt{\pi}\frac{\frac{(2d')!}{4^{d'}d'!}}{(d'-1)!} = \sqrt{\pi}d'4^{-d'}\binom{2d'}{d'} \tag{29}$$

$$\approx \sqrt{\pi}d'4^{-d'}\left( \frac{4^{d'}}{\sqrt{\pi d'}} + 4^{d'}O(d'^{-1.5}) \right) = \sqrt{d'} + O(d'^{-0.5}) = \sqrt{\frac{d}{2}} + O(d^{-0.5}), \tag{30}$$

where the approximation of the central binomial coefficient holds asymptotically (Luke, 1969, p. 35). $\square$

## C   DETAILS ON THE DISTANCES

Let $x_1, \ldots, x_n \in \mathbb{R}^d$. We denote pairwise Euclidean distances by $d_{ij} = \|x_i - x_j\|$, the $k$ nearest neighbors of $x_i$ in increasing distance by $x_{i_1}, \ldots, x_{i_k}$, and the set containing them by $N_i$. Many distances rely on the symmetric $k$-nearest-neighbor (s$k$NN) graph. This graph contains edge $ij$ if $x_i$ is among the $k$ nearest neighbors of $x_j$ or vice versa.

**Fermat distances**   For $p \geq 1$, the Fermat distance is defined as

$$d_p^F(i,j) = \inf_\pi \left( \sum_{(uv) \in \pi} d_{uv}^p \right),$$
(31)

where the infimum is taken over all finite paths from $x_i$ to $x_j$ in the complete graph with edge weights $d_{ij}^p$. As a speed-up, Fernández et al. (2023) suggested to compute the shortest paths only on the $k$NN graph, but for our sample sizes we could perform the calculation on the complete graph. For $p = 1$ this reduces to normal Euclidean distances due to the triangle inequality. We used $p \in \{2, 3, 5, 7\}$.

**DTM distances**   The DTM distances depend on three hyperparameters: the number of nearest neighbors $k$, one hyperparameter controlling the distance to measure $p$, and finally a hyperparameter $\xi$ controlling the combination of DTM and Euclidean distance. The DTM value for each point is given by

$$\mathrm{dtm}_i = \begin{cases} \sqrt[p]{\sum_{\kappa=1}^k \|x_i - x_{i_\kappa}\|^p / k} & \text{if } p < \infty \\ \|x_i - x_{i_k}\| & \text{else.} \end{cases}$$
(32)

These values are combined with pairwise Euclidean distances to give pairwise DTM distances:

$$d_{k,p,\xi}^{\mathrm{DTM}}(i,j) = \begin{cases} \max(\mathrm{dtm}_i, \mathrm{dtm}_j) & \text{if } \|x_i - x_j\| \leq \sqrt[\xi]{|\mathrm{dtm}_i^\xi - \mathrm{dtm}_j^\xi|} \\ \theta & \text{else,} \end{cases}$$
(33)

where $\theta$ is the only positive root of $\sqrt[\xi]{\theta^\xi - \mathrm{dtm}_i^\xi} + \sqrt[\xi]{\theta^\xi - \mathrm{dtm}_j^\xi} = d_{ij}$. We only considered the values $\xi \in \{1, 2, \infty\}$, for which the there are closed-form solutions:

$$\theta = \begin{cases} (\mathrm{dtm}_i + \mathrm{dtm}_j + d_{ij})/2 & \text{if } \xi = 1 \\ \sqrt{((\mathrm{dtm}_i + \mathrm{dtm}_j)^2 + d_{ij}^2) \cdot ((\mathrm{dtm}_i - \mathrm{dtm}_j)^2 + d_{ij}^2)}/(2d_{ij}) & \text{if } \xi = 2 \\ \max(\mathrm{dtm}_i, \mathrm{dtm}_j, d_{ij}/2) & \text{if } \xi = \infty. \end{cases}$$
(34)

We used $k \in \{4, 15, 100\}$, $p \in \{2, \infty\}$, and $\xi \in \{1, 2, \infty\}$.

The original exposition of DTM-based filtrations Anai et al. (2020) only considered the setting $\xi = 2$, while DTM has been defined for arbitrary $p \geq 1$ Chazal & Michel (2021). We explore an additional value, $p = \infty$, in order to possibly strengthen DTM. Indeed, in several experiments it outperformed the $p = 2$ setting.

Moreover, Anai et al. (2020) actually used a small variant of the Vietoris-Rips complex on the above distance $d_{ij}^{\mathrm{DTM}}(k, p, \xi)$: They only included point $x_i$ in the filtered complex once the filtration value exceeds $\mathrm{dtm}_i$. This, however, only affects the 0-th homology, which we do not consider in our experiments.

**Core distance**   The core distance is similar to the DTM distance with $\xi = \infty$ and $p = \infty$ and is given by

$$d_k^{\mathrm{core}}(i,j) = \max(d_{ij}, \|x_i - x_{i_k}\|, \|x_j - x_{j_k}\|).$$
(35)

We used $k \in \{15, 100\}$.

**$t$-SNE graph affinities**   The $t$-SNE affinities are given by

$$p_{ij} = \frac{p_{i|j} + p_{j|i}}{2n}, \quad p_{j|i} = \frac{\nu_{j|i}}{\sum_{k \neq i} \nu_{k|i}}, \quad \nu_{j|i} = \begin{cases} \exp\left( \|x_i - x_j\|^2 / (2\sigma_i^2) \right) & \text{if } x_j \in N_i \\ 0 & \text{else,} \end{cases}$$
(36)

where $\sigma_i$ is selected such that the distribution $p_{j|i}$ has pre-specified perplexity $\rho$. Standard implementations of $t$-SNE use $k = 3\rho$. We transformed $t$-SNE affinities into pairwise distances by taking the negative logarithm. Pairs $x_i$ and $x_j$ with $p_{ij} = 0$ (i.e. not in the $k$NN graph) get distance $\infty$. We used $\rho \in \{30, 200, 333\}$.

**UMAP graph affinities**   The UMAP affinities are given by

$$\mu_{ij} = \mu_{i|j} + \mu_{j|i} - \mu_{i|j}\mu_{j|i}, \quad \mu_{j|i} = \begin{cases} \exp\big(-(d_{ij} - \mu_i)/\sigma_i\big) & \text{for } j \in \{i_1, \ldots, i_k\} \\ 0 & \text{else,} \end{cases} \tag{37}$$

where $\mu_i = \|x_i - x_{i_1}\|$ is the distance between $x_i$ and its nearest non-identical neighbor. The scale parameter $\sigma_i$ is selected such that

$$\sum_{\kappa=1}^{k} \exp\Big(-\big(d(x_i, x_{i_\kappa}) - \mu_i\big)/\sigma_i\Big) = \log_2(k). \tag{38}$$

As above, to convert these affinities into distances, we take the negative logarithm and handle zero similarities as for the $t$-SNE case. We used $k \in \{100, 999\}$; $k = 15$ resulted in memory overflow on one of the void-containing datasets.

Gardner et al. (2022) and Hermansen et al. (2022) first used these distances, but omitted $\mu_i$, which we included to completely reproduce UMAP's affinities.

Note that distances derived from UMAP and $t$-SNE affinities are not guaranteed to obey the triangle inequality.

**Geodesic distances**   We computed the shortest path distances between all pairs of nodes in the s$k$NN graph with edges weighted by their Euclidean distances. We used the python function `scipy.sparse.csgraph.shortest_path`. We used $k = \{15, 100\}$.

**UMAP embedding**   We computed the UMAP embeddings in 2 embedding dimensions using 750 optimization epochs, `min_dist` of 0.1, exactly computed $k$ nearest neighbors, and PCA initialization. Then we used Euclidean distances between the embedding points. We used UMAP commit `a7606f2`. We used $k \in \{15, 100, 999\}$.

$t$**-SNE embedding**   We computed the $t$-SNE embeddings in 2 embedding dimensions using openTSNE (Poličar et al., 2019) with default parameters, but providing manually computed affinities. For that we used standard Gaussian affinities on the s$k$NN graph with $k = 3\rho$. Then we used the Euclidean distances between the embedding points. We used perplexity $\rho \in \{8, 30, 333\}$.

For UMAP and $t$-SNE affinities as well as for UMAP and $t$-SNE embeddings we computed the s$k$NN graph with PyKeOps (Charlier et al., 2021) instead of using the default approximate methods. The UMAP and $t$-SNE affinities (without negative logarithm) were used by the corresponding embedding methods.

**Effective resistance**   We computed the effective resistance on the s$k$NNgraph. Following the analogy with resistances in an electric circuit, if the s$k$NN graph is disconnected, we computed the effective resistance separately in each connected component and set resistances between components to $\infty$. The uncorrected resistances were computed via the pseudoinverse of the graph Laplacian

$$\tilde{d}_{ij}^{\text{eff}} = l_{ii}^\dagger - 2l_{ij}^\dagger + l_{jj}^\dagger, \tag{39}$$

where $l_{ij}^\dagger$ is the $ij$-th entry of the pseudoinverse of the non-normalized s$k$NN graph Laplacian. For the corrected version, we used

$$d_{ij}^{\text{eff}} = \tilde{d}_{ij}^{\text{eff}} - \frac{1}{d_i} - \frac{1}{d_j} + 2\frac{a_{ij}}{d_i d_j} - \frac{a_{ii}}{d_i^2} - \frac{a_{jj}}{d_j^2}. \tag{40}$$

For the weighted version of effective resistance, each edge in the s$k$NN graph was weighted by the inverse of the Euclidean distance. We experimented with the weighted and unweighted versions, but only reported the unweighted version in the paper (as the difference was always minor). We also experimented with the unweighted and uncorrected version and saw that correcting is crucial for high noise levels (Figure S7). We used $k \in \{15, 100\}$.

**Diffusion distance**   We computed the diffusion distances on the unweighted s$k$NN graph directly by equation (2), i.e.,

$$d_t^{\text{diff}}(i, j) = \sqrt{\text{vol}(G)}\|(P_{i,:}^t - P_{j,:}^t)D^{-1}\|. \tag{41}$$

Note that our s$k$NN graphs do not contain self-loops. We used $k \in \{15, 100\}$ and $t \in \{8, 64\}$.

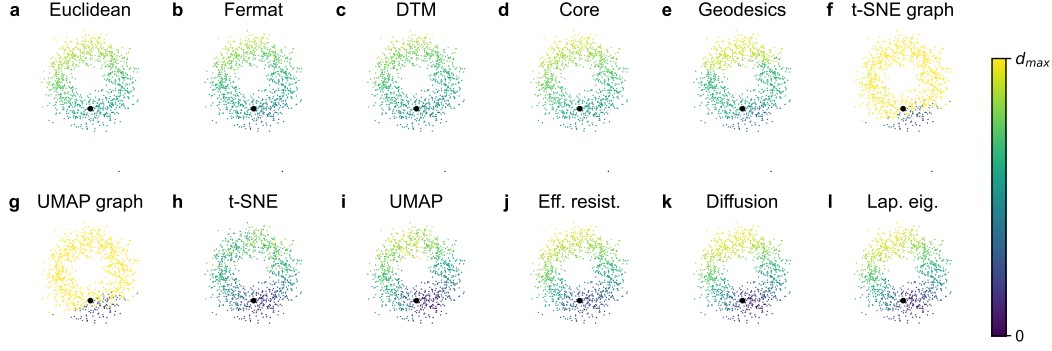

Figure S1: Visualization of all distances on the noisy circle in $\mathbb{R}^{50}$ with $\sigma = 0.25$. The scatter plots are all the 2D PCA of the 50D dataset. The colors indicate the distance to the highlighted point.

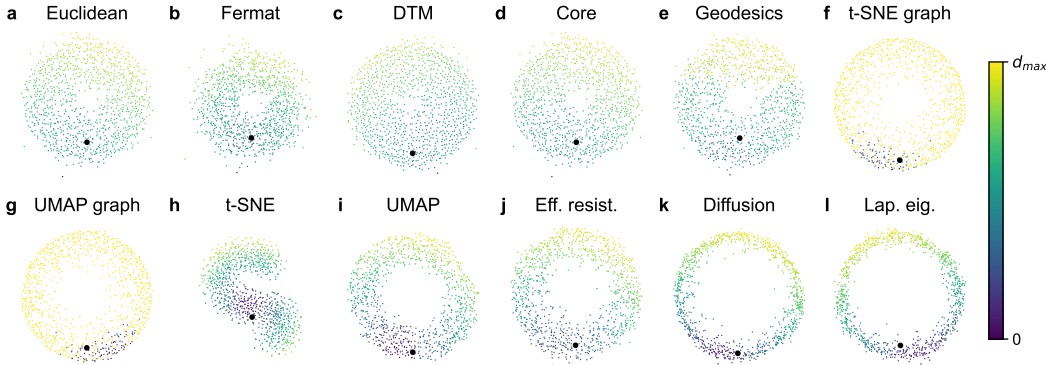

Figure S2: Visualization of all distances on the noisy circle in $\mathbb{R}^{50}$ with $\sigma = 0.25$. The scatter plots are 2D multidimensional scaling embeddings using the respective distances. The colors indicate the distance to the highlighted point. For this seed the $t$-SNE embedding tore the circle apart.

**Laplacian Eigenmaps** For a dataset with $K$ connected components, we computed the $K + \tilde{d}$ eigenvectors $v_1, \ldots, v_{K+\tilde{d}}$ of the un-normalized graph Laplacian of the $sk$NN graph and discarded the first $K$ eigenvectors $v_1, \ldots, v_K$, which are just indicators for the connected components. Then we computed the Euclidean distances between the embedding vectors $e_i^{\text{LE}} = (v_{K+1,i}, \ldots, v_{(K+\tilde{d}),i})$. We used $k = 15$ and embedding dimensions $\tilde{d} \in \{2, 5, 10\}$.

Alternatively, one can compute Laplacian Eigenmaps using the normalized graph Laplacian $L^{\text{sym}} = D^{-\frac{1}{2}} L D^{-\frac{1}{2}}$. We tried this normalization for $\tilde{d} = 2$ but obtained very similar embeddings.

For all methods, we replaced infinite distances with twice the maximal finite distance to be able to compute our hole detection scores.

# D DATASETS

## D.1 SYNTHETIC DATASETS

The synthetic, noiseless datasets with $n = 1\,000$ points each are depicted in Figure S3 noised versions of the circle for ambient dimensions $d = 2, 50$ are depicted in Figure S4.

**Circle** The circle dataset consists of $n$ points equidistantly spaced along a circle of radius $r = 1$.

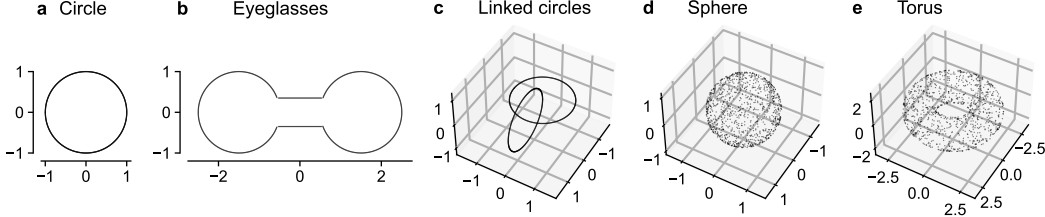

Figure S3: Synthetic, noiseless datasets with $n = 1\,000$ points each.

**Linked circles**   The linked circles dataset consists of two circle datasets of $n/2$ points each, arranged such that each circle perpendicularly intersects the plane spanned by the other.

**Eyeglasses**   The eyeglasses dataset consists of four parts: Two circle segments of arclength $\pi + 2.4$ and radius $r = 1$, centered 3 units apart with the gaps facing each other. The third and fourth part are two straight line segments of length $1.06$, separated by $0.7$ units linking up the two circle segments. The circle segments consist of $0.425n$ equidistantly distributed points each and the line segments consist of $0.075n$ equispaced points each. As the length scale of this dataset is dominated by the bottleneck between the two line segments, we only considered noise levels $\sigma \in [0, 0.15]$ for this dataset, as at this point the bottleneck essentially merges in $\mathbb{R}^2$.

**Sphere**   The sphere dataset consists of $n$ points sampled uniformly from a sphere $S^2$ with radius $r = 1$.

**Torus**   The torus dataset consists of $n$ points sampled uniformly from a torus. The radius of the torus' tube was $r = 1$ and the radius of the center of the tube was $R = 2$. Note that we do not sample the points to have uniform angle distribution along the tube's and the tube center's circle, but uniform on the surface of the torus.

**High-dimensional noise**   We mapped each dataset to $\mathbb{R}^d$ for $d \in [2, 50]$ using a random matrix $\mathbf{V}$ of size $d \times 2$ or $d \times 3$ with orthonormal columns, and then added isotropic Gaussian noise sampled from $\mathcal{N}(\mathbf{0}, \sigma^2 \mathbf{I}_d)$ for $\sigma \in [0, 0.35]$.

The orthogonal embedding in $\mathbb{R}^d$ does not change the shape of the data. The procedure is equivalent to adding $d - 2$ or $d - 3$ zero dimensions and then randomly rotating the resulting dataset in $\mathbb{R}^d$.

### D.2   SINGLE-CELL DATASETS

We depict 2D embeddings of all single-cell datasets in Figure S5.

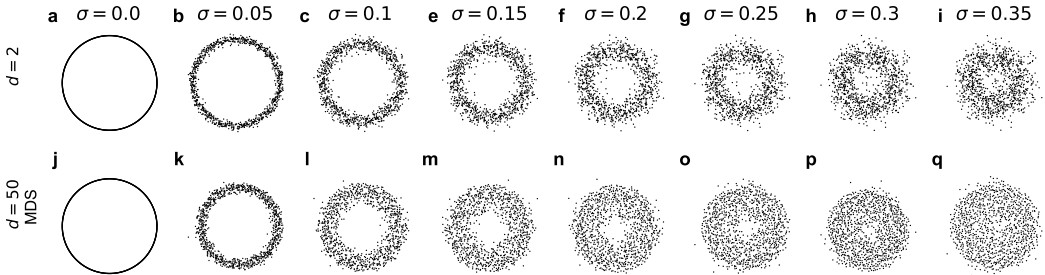

Figure S4: Circle with Gaussian noise of different standard deviation $\sigma$. **a−i.** Original data in ambient dimension $d = 2$. **j−q.** Multidimensional scaling of the Euclidean distance of the data in ambient dimension $d = 50$.

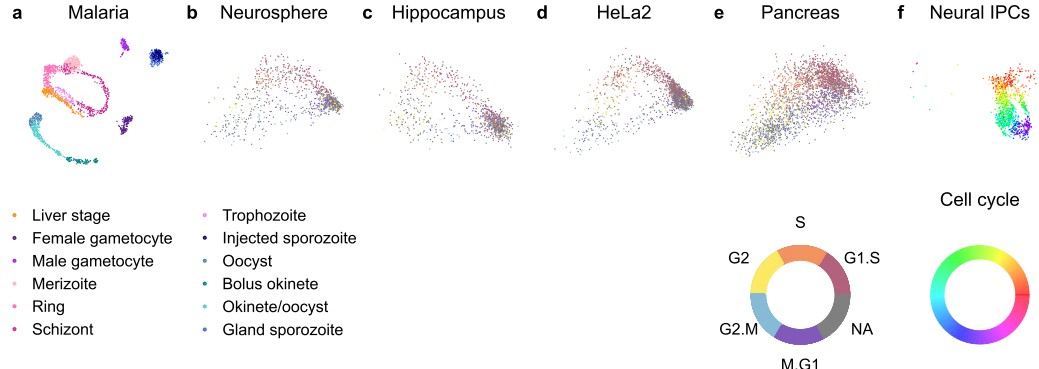

Figure S5: 2D embeddings of all six single-cell datasets. **a, f.** UMAP embeddings of the Malaria (Howick et al., 2019) and the Neural IPC datasets (Braun et al., 2022). We recomputed the embedding for the Malaria dataset using UMAP hyperparameters provided in the original publication, and subsetted a provided UMAP of a superset of telencephalic exitatory cells to the Neural IPC. Text legend refers to Malaria cell types. **b – e.** 2D linear projection constructed to bring out the cell cycle ('tricycle embedding') (Zheng et al., 2022) of the Neurosphere, Hippocampus, HeLa2, and Pancreas datasets. We used the projection coordinates provided by Zheng et al. (2022).

**Malaria**  The Malaria dataset (Howick et al., 2019) consists of gene expression measurement of $5\,126$ genes obtained with the modified SmartSeq2 approach of Reid et al. (2018) in $n = 1\,787$ cells from the entire life cycle of Plasmodium berghei. The resulting transcripts were pre-processed with the trimmed mean of M-values method (Robinson & Oshlack, 2010). We obtained the pre-processed data from `https://github.com/vhowick/MalariaCellAtlas/raw/v1.0/Expression_Matrices/Smartseq2/SS2_tmmlogcounts.csv.zip`. The UMAP embedding shown in Figure 8 follows the authors' setup and uses correlation distance as input metric, $k = 10$ nearest neighbors, and a `min_dist` of 1 and `spread` of 2. Note that when computing persistent homology with UMAP-related distances, we used our normal UMAP hyperparameters and never changed `min_dist` or `spread`.

**Neural IPCs**  The Neural IPC dataset (Braun et al., 2022) consists of gene expressions of $n = 26\,625$ neural IPCs from the developing human cortex. scVI (Lopez et al., 2018) was used to integrate cells with different ages and donors based on the 700 most highly variable genes, resulting in a $d = 10$ dimensional embedding. Braun et al. (2022) shared this representation with us for a superset of 297 927 telencephalic exitatory cells. We limited our analysis to the neural IPCs because they formed a particularly prominent cell cycle.

**Neurosphere**  The Neurosphere dataset (Zheng et al., 2022) consists of gene expressions for $n = 12\,805$ cells from the mouse neurosphere. After quality control, the data was library size normalized and $\log_2$ transformed. Seurat was used to integrate different samples based on the first 30 PCs of the top $2\,000$ highly variable genes, resulting in a $12\,805 \times 2\,000$ matrix of $\log_2$ transformed expressions.These were subsetted to the genes in the gene ontology (GO) term cell cycle (GO:0007049). The $500$ most highly variable genes are selected and a PCA was computed to $d = 20$. The GO PCA representation was downloaded from `https://zenodo.org/record/5519841/files/neurosphere.qs`.

**Hippocampus**  The Hippocampus dataset (Zheng et al., 2022) consists of gene expressions for $n = 9\,188$ mouse hippocampal NPCs. The pre-processing was the same as for the Neurosphere dataset. The GO PCA representation was downloaded from `https://zenodo.org/record/5519841/files/hipp.qs`.

**HeLa2**  The HeLa2 dataset (Schwabe et al., 2020; Zheng et al., 2022) consists of gene expressions for $2\,463$ cells from a human cell line derived from cervical cancer. After quality control, the data was library size normalized and $\log_2$ transformed. From here the GO PCA computation was the

Table S1: The optimal hyperparameters that were selected in Figure 6. For torus and sphere, we consider the case of loop detection ($H_1$) and void detection ($H_2$) separately.

| Dataset | Fermat | DTM | Eff. res. | Diffusion |
|---------|--------|-----|-----------|-----------|
| Circle | $p = 3$ | $k = 4, p = 2, \xi = 1$ | $k = 100$ | $k = 100, t = 8$ |
| Eyeglasses | $p = 7$ | $k = 100, p = 2, \xi = 1$ | $k = 15$ | $k = 15, t = 64$ |
| Linked circles | $p = 7$ | $k = 15, p = \infty, \xi = 1$ | $k = 15$ | $k = 15, t = 8$ |
| Torus $H_1$ | $p = 2$ | $k = 4, p = 2, \xi = \infty$ | $k = 100$ | $k = 15, t = 8$ |
| Sphere $H_1$ | $p = 2$ | $k = 100, p = 2, \xi = 1$ | $k = 15$ | $k = 100, t = 64$ |
| Torus $H_2$ | $p = 2$ | $k = 4, p = 2, \xi = \infty$ | $k = 100$ | $k = 15, t = 8$ |
| Sphere $H_2$ | $p = 2$ | $k = 4, p = 2, \xi = 1$ | $k = 100$ | $k = 100, t = 8$ |

same as for the neurosphere dataset. The GO PCA representation was downloaded from `https://zenodo.org/record/5519841/files/HeLa2.qs`.

**Pancreas** The Pancreas dataset (Bastidas-Ponce et al., 2019; Zheng et al., 2022) consists of gene expressions for $3\,559$ cells from the mouse endocrine pancreas. After quality control, the data was library size normalized and $\log_2$ transformed. From here the GO PCA computation was the same as for the neurosphere dataset. The GO PCA representation was downloaded from `https://zenodo.org/record/5519841/files/endo.qs`.

## E  HYPERPARAMETER SELECTION

For each of the datasets and hole dimensions, we showed the result with the best hyperparameter setting. For the synthetic experiments, this meant the highest area under the hole detection score curve, while for the single-cell datasets it meant the highest loop detection score. Here, we give details of the selected hyperparamters.

For Figure 1 we used effective resistance with $k = 100$ as in Figure 5.

For Figure 5 we specified the selected hyperparameters directly in the figure. For the density-based methods, they were $p = 3$ for Fermat distances, $k = 4, p = 2, \xi = \infty$ for DTM, and $k = 15$ for the core distance. For the graph-based methods, they were $k = 100$ for the geodesics, $k = 100$ for the UMAP graph affinities, and $\rho = 30$ for $t$-SNE graph affinities. The embedding-based methods used $k = 15$ for UMAP and $\rho = 30$ for $t$-SNE. Finally, as spectral methods, we selected effective resistance with $k = 100$, diffusion distance with $k = 15, t = 8$ and Laplacian Eigenmaps with $k = 15, d = 2$.

The hyperparameters for Figure 6 are given in Table S1.

In Figure 7 we specified the hyperparameters used. They were the same as for Figure 5.

For Figure 8, we selected DTM with $k = 15, p = \infty, \xi = \infty$, effective resistance with $k = 15$ and diffusion distance with $k = 15, t = 64$. They are the same for the Malaria dataset in Figure 9.

The selected hyperparameters for Figure 9 can be found in Table S2.

The hyperparameters for Figures S1 and S2 are the same as those used in Figure 5.

## F  IMPLEMENTATION DETAILS

We computed persistent homology using the `ripser` (Bauer, 2021) project's `representative-cycles` branch at commit `140670f` to compute persistent homologies and representative cycles. We used coefficients in $\mathbb{Z}/2\mathbb{Z}$. To compute $k$NN graphs, we used the PyKeops package (Charlier et al., 2021). The rest of our implementation is in python. Our code is available in the supplementary material.

Our experiments were run on a machine with a Intel(R) Xeon(R) Gold 6226R CPU @ 2.90GHz with 64 kernels and an NVIDIA RTX A6000 GPU.

Table S2: The optimal hyperparameters that were selected in Figure 9. For DTM we report the best setting without thresholding (because none of the DTM runs passed our birth/death thresholding, so all $s_m$ scores for all parameter combinations are zero).

| Dataset | Fermat | DTM | $t$-SNE | UMAP | Eff. res. | Diffusion | Lap. Eig. |
|---|---|---|---|---|---|---|---|
| Malaria | $p = 2$ | $k = 15$ $p = \infty$ $\xi = \infty$ | $\rho = 8$ | $k = 15$ | $k = 15$ | $k = 15$ $t = 64$ | $D = 10$ |
| Neurosphere | $p = 2$ | $k = 100$ $p = 2$ $\xi = 2$ | $\rho = 30$ | $k = 999$ | $k = 15$ | $k = 15$ $t = 8$ | $D = 2$ |
| Hippocampus | $p = 7$ | $k = 100$ $p = 2$ $\xi = 2$ | $\rho = 8$ | $k = 15$ | $k = 100$ | $k = 15$ $t = 8$ | $D = 10$ |
| Neural IPC | $p = 2$ | $k = 4$ $p = \infty$ $\xi = \infty$ | $\rho = 30$ | $k = 15$ | $k = 100$ | $k = 15$ $t = 8$ | $D = 2$ |
| HeLa2 | $p = 3$ | $k = 4$ $p = 2$ $\xi = \infty$ | $\rho = 30$ | $k = 100$ | $k = 100$ | $k = 100$ $t = 8$ | $D = 2$ |
| Pancreas | $p = 7$ | $k = 100$ $p = 2$ $\xi = \infty$ | $\rho = 8$ | $k = 100$ | $k = 4$ | $k = 15$ $t = 64$ | $D = 2$ |

Table S3: Exemplary run times in seconds.

| Dataset | n | $\sigma$ | Distance | Feature dim | Time distance [s] | Time PH [s] |
|---|---|---|---|---|---|---|
| Circle | 1 000 | 0.0 | Euclidean | 1 | $0.013 \pm 0.002$ | $12.3 \pm 0.4$ |
| Circle | 1 000 | 0.0 | Eff. res $k = 100$ | 1 | $0.17 \pm 0.04$ | $12.0 \pm 0.2$ |
| Circle | 2 000 | 0.0 | Euclidean | 1 | $0.09 \pm 0.04$ | $117 \pm 9$ |
| Sphere | 1 000 | 0.0 | Euclidean | 1 | $0.012 \pm 0.001$ | $1.31 \pm 0.06$ |
| Sphere | 1 000 | 0.0 | Euclidean | 2 | $0.017 \pm 0.002$ | $4687 \pm 2501$ |
| Circle | 1 000 | 0.35 | Euclidean | 1 | $0.016 \pm 0.001$ | $5 \pm 2$ |
| Sphere | 1 000 | 0.35 | Euclidean | 2 | $0.03 \pm 0.02$ | $258 \pm 18$ |

Our benchmark consisted of many individual experiments. We explored 47 hyperparameter settings across all distances, computed results for 3 random seeds and 29 noise levels $\sigma$. In the synthetic benchmark, we computed only 1D persistent homology for 3 datasets and both 1D and 2D persistent homology of 2 more datasets. So the synthetic benchmark with ambient dimension $d = 50$ alone consisted of 12 267 computations of 1D persistent homology and 8 178 computations of both 1D and 2D persistent homology.

The run time of persistent homology vastly dominated the time taken by the distance computation. The persistent homology run time depended most strongly on the sample size $n$, the dataset, and on the highest dimensionality of holes. The difference between distances was usually small. However, we observed that there were some outliers, depending on the noise level and the random seed, that had much longer run time. Overall, we found that methods that produce many pairwise distances of the same value (e.g., because of infinite distance in the graph affinities or maximum operations like for DTM with $p = \infty, \xi = \infty$) often had a much longer run time than other settings. We presume this was because equal distances lead to many simplices being added to the complex at the same time. We give exemplary run times in Table S3.

As a rough estimate for the total run time, we extrapolated the run times for the circle to all 1D persistent homology experiments for ambient dimension $d = 50$ and the times for the sphere to all 2D experiments. In both cases we took the mean between the noiseless ($\sigma = 0$) and highest noise ($\sigma = 0.35$) setting in Table S3. This way, we estimated a total sequential run time of about 57 days, but we parallelized the runs.

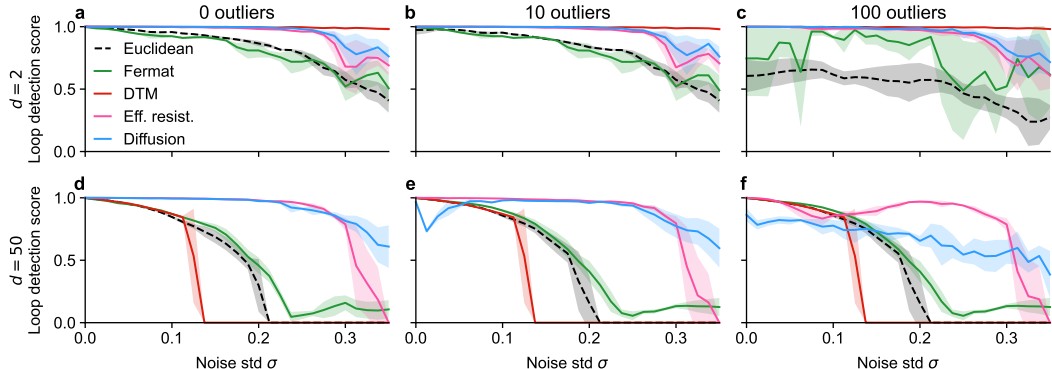

Figure S6: Loop detection performance of various methods on the noisy circle in the presence of outliers in low- and high-dimensional ambient space. Outliers are sampled uniformly from an axis-aligned cube around the data. **a – c.** In low ambient dimension ($d = 2$) adding 100 outliers hurt the performance of the Euclidean distance, but barely that of the spectral methods and not at all DTM's excellent performance. **d – f.** In high ambient dimension ($d = 50$) all methods were fairly robust to outliers, leading to effective resistance and diffusion distance outperforming the others.

## G   EFFECT OF OUTLIERS

Persistent homology with the Euclidean distance is known to be sensitive to outliers. Methods such as DTM were introduced to handle this issue. However, spectral methods can handle outliers decently, too. Moreover, outliers are very sparse in high ambient dimension making them less of a problem. We experimented with the noisy circle with $n = 1\,000$ points in ambient $\mathbb{R}^d$ for $d = 2, 50$ and added either 10 or 100 outlier points. These were sampled uniformly from axis-aligned cubes around the data in ambient space. The size of the cube was set just large enough that it contains the data even with the strongest Gaussian noise added. In low dimension, Euclidean distance suffered severely for 100 outliers and Fermat distances led to high uncertainty in this setting. Diffusion distance and effective resistance were much more outlier-resistant than the Euclidean distance and changed their performance barely. DTM excelled in this setting, being completely insensitive to outliers and achieving top score for all noise levels. Adding only 10 outliers was unproblematic of all methods (Figure S6 a – c).

The volume of the bounding box in $d = 50$ ambient dimensions is much larger and thus the same number outlier points lie much more sparsely. In particular, it is much less likely that an outlier happens to fall into the middle of the circle. As a result, even Euclidean and Fermat distance were very outlier robust in $d = 50$ ambient dimensions (Figure S6 d – f). Similarly, DTM's performance did not change at all in the face of outliers. However, all three methods suffered from the high-dimensional Gaussian noise. Diffusion distance's performance dipped for 10 outliers in the low noise setting and deteriorated overall for 100, while still performing well in the high-noise regime. Effective resistance performed best, deteriorating only slightly for 100 outliers in the low noise setting.

To sum up, effective resistance (and to a lesser extent diffusion distance) can handle both outliers and high-dimensional Gaussian noise, while other methods can handle at most one type of noise.

# H    ADDITIONAL FIGURES

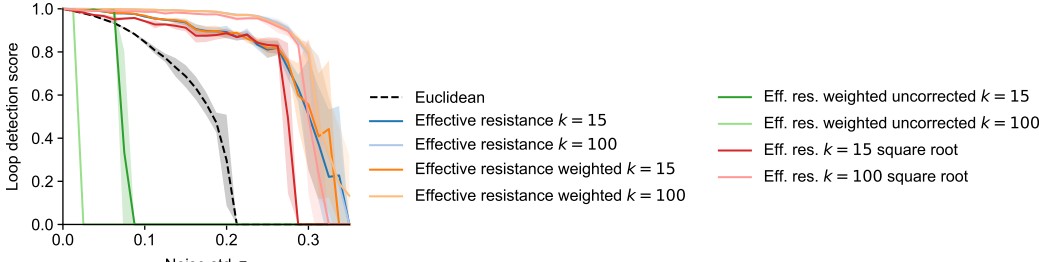

Figure S7: Loop detection score on noisy $S^1 \subset \mathbb{R}^{50}$ for various versions of effective resistance. There was little difference between using the weighted $k$NN graph, unweighted $k$NN graph, and using the square root of effective resistance based on the unweighted $k$NN graph. The latter got filtered out for high noise levels. Using $k = 100$ instead of $k = 15$ helped only marginally on this dataset. The uncorrected (naive) version of effective resistance collapsed at very small noise levels.

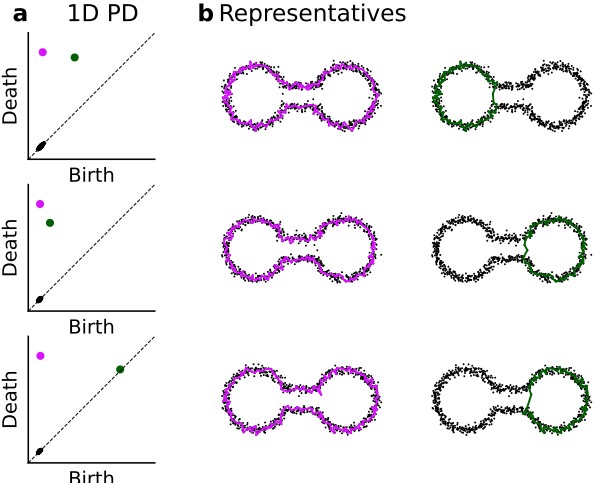

Figure S8: Illustration for the uncertainty of effective resistance on the noisy eyeglasses dataset in $\mathbb{R}^{50}$ with $\sigma = 0.075$. This refers to panel b of Fig. 6. **a** One dimensional persistence diagrams for three random seeds. **b.** Representatives of the most two most persistent features superimposed on a 2D PCA of the dataset. These always corresponded to the full shape and one of the two circle segments. For the first two random seeds, some points are distorted in such a way that they form a bridge in the 2D PCA, while in the third there is not such bridge and the second most persistent feature is much less persistent. Note that this is just a 2D PCA, in particular, much of the noise in 50D is not visible. A similar explanation applies for the diffusion distance in panel b of Fig. 6.

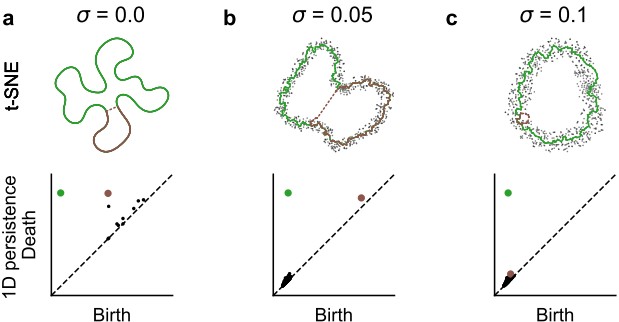

Figure S9: $t$-SNE embeddings and 1D persistence diagrams of the embedding for a circle in ambient $\mathbb{R}^{50}$ with Gaussian noise of low standard deviation $\sigma$. Perplexity $\rho = 8$ is rather small, such that each embedding point only has attraction to very few other points. In the noiseless setting this very sparse attraction is only among immediate neighbors along the circle. This allows the embedding to have spurious curves. For higher noise, the sparse attraction pattern is less regular and local such that the spurious curves disappear. The more spurious curves the embedding has, the more high persistent features, given by bottlenecks in the curvy embedding, exist. This explains the dip for the $t$-SNE $\rho = 8$ curve in panel g of Fig. S10.

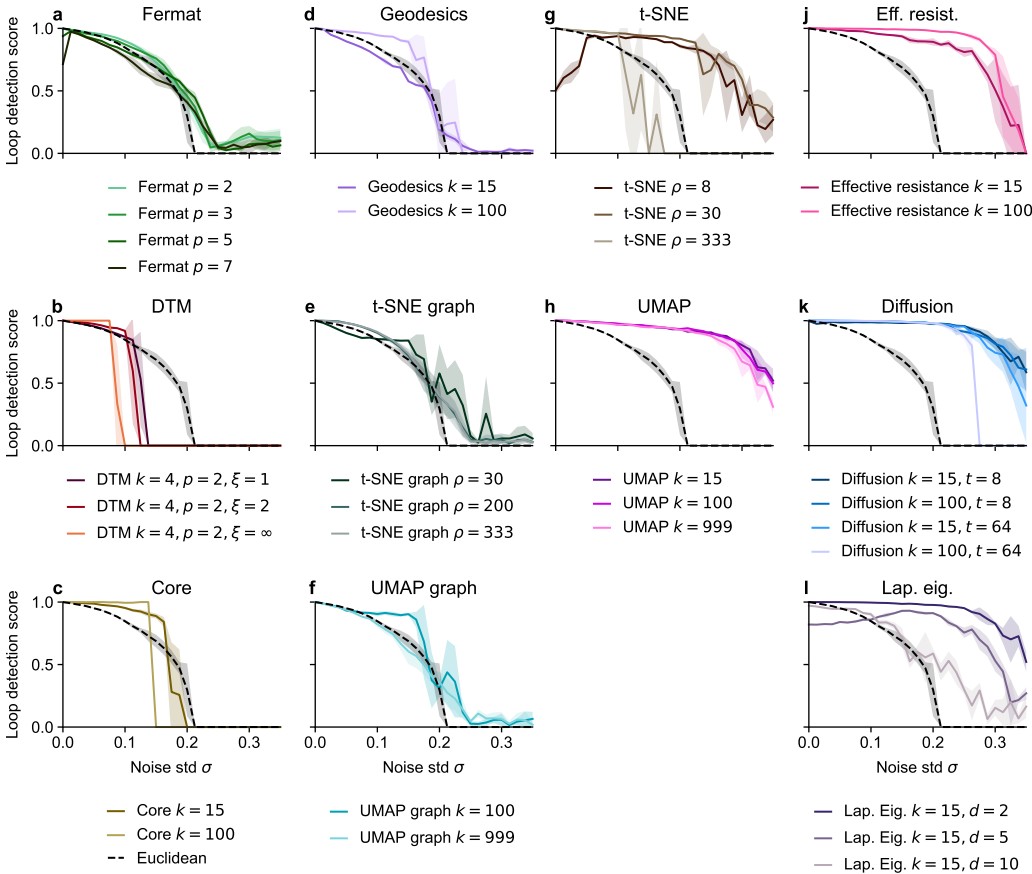

Figure S10: Loop detection score for persistent homology with various distances on a noisy circle in ambient $\mathbb{R}^{50}$. Extension of Figure 5. Spectral and embedding methods perform best. The reason for the dip for the low-perplexity $t$-SNE embedding is depicted in Figure S9.

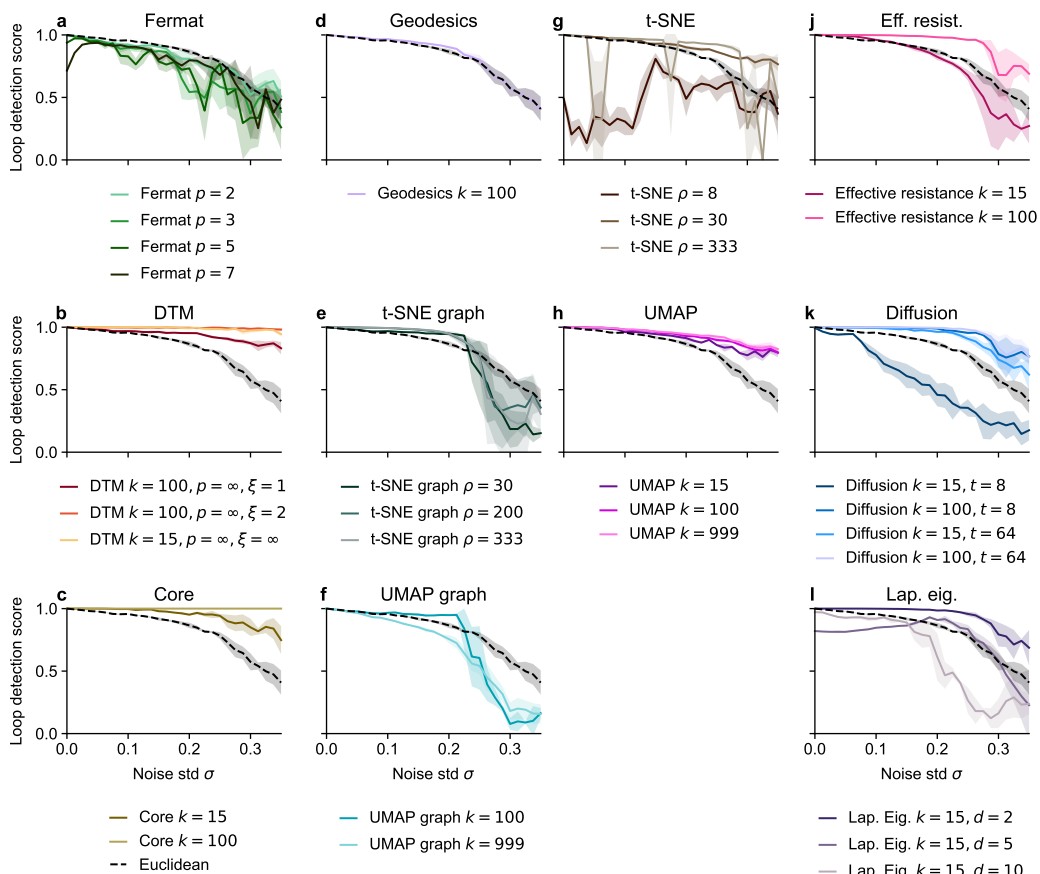

Figure S11: Loop detection score for persistent homology with various distances on a noisy circle in ambient $\mathbb{R}^2$. Our code for finding the geodesics for $k = 15$ did not terminate. Nearly all methods perform near perfects for most noise levels. Note the striking difference to the 50D setting in Fig. S10

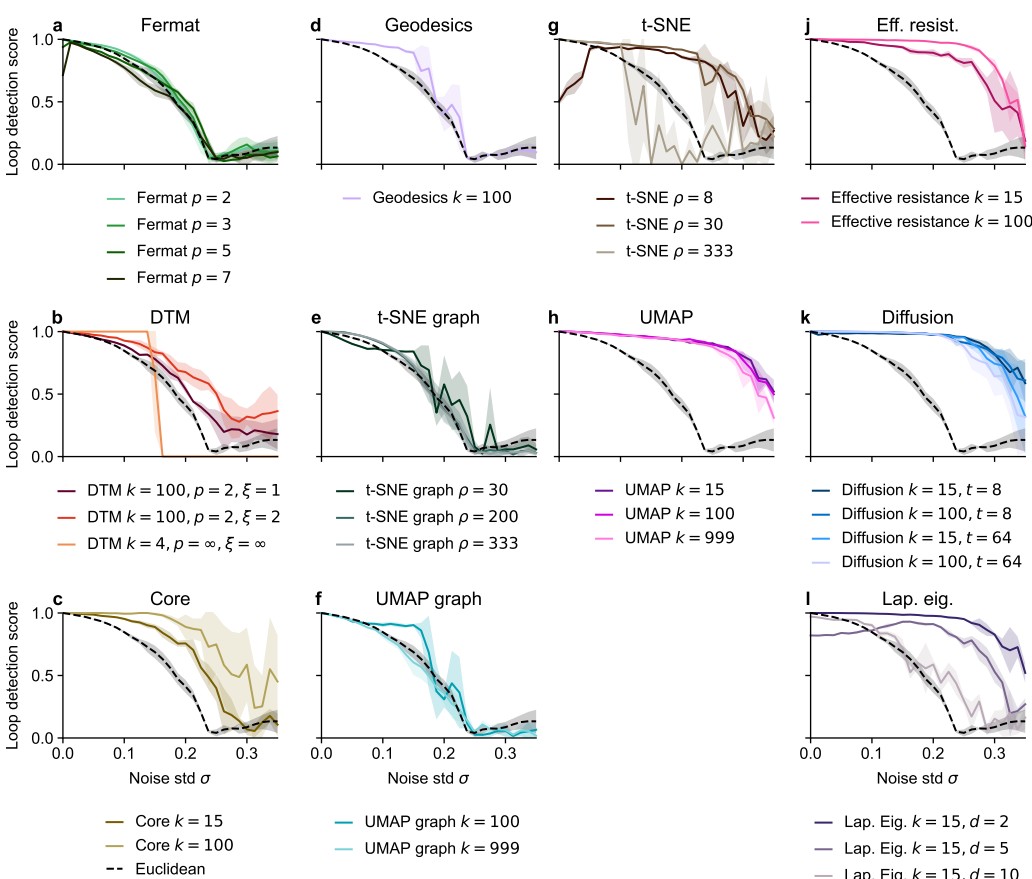

Figure S12: Loop detection score for persistent homology with various distances on a noisy circle in ambient $\mathbb{R}^{50}$. No thresholding was used for this figure, in contrast to Fig. S10. Without thresholding, DTM had better performance, but not much beyond the level of Euclidean distance. Several issues such as high uncertainty for Core $k = 100$, $t$-SNE $\rho = 333$ and artifactual increasing performance for several methods at very high noise levels are not handled anymore.

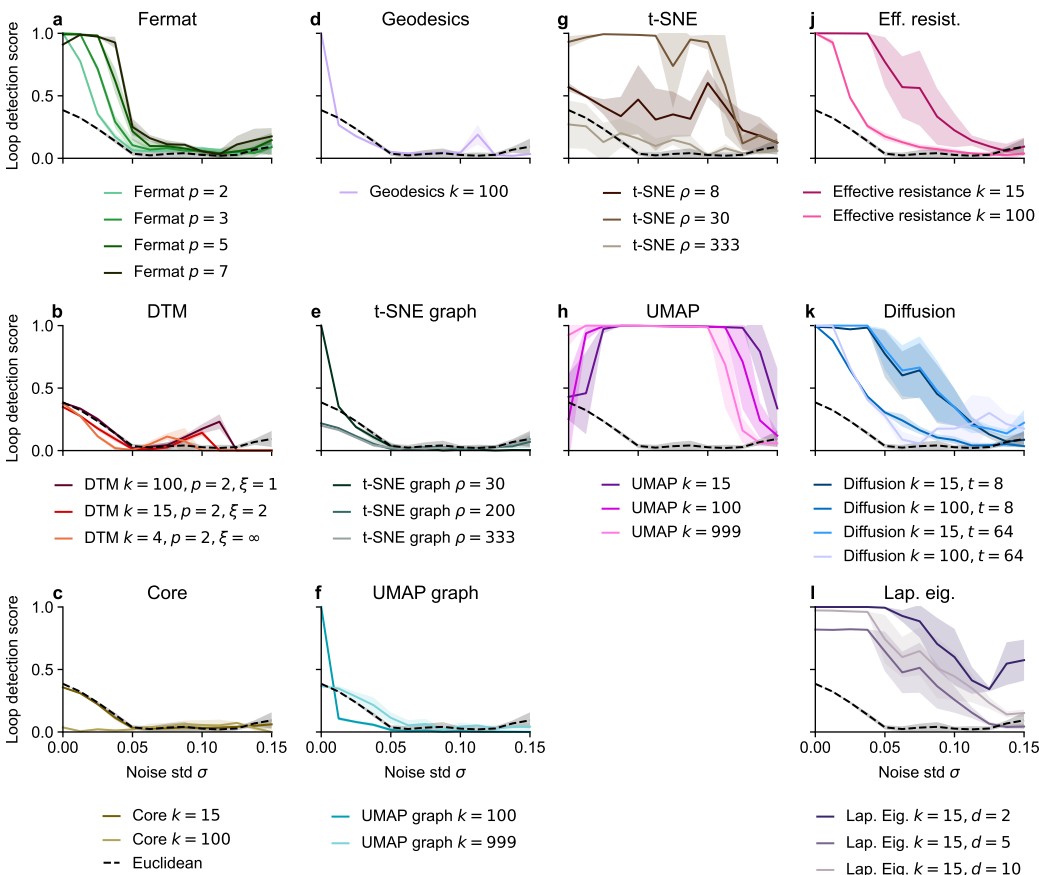

Figure S13: Loop detection score for persistent homology with various distances on the noisy eye-glasses dataset in ambient $\mathbb{R}^{50}$. Only Fermat distance, embedding methods and spectral method could handle noise low noise levels. Some embedding methods performed very well, but UMAP struggles in the low noise setting. The reason for the high uncertainty for effective resistance with $k = 15$ is depicted in Figure S8.

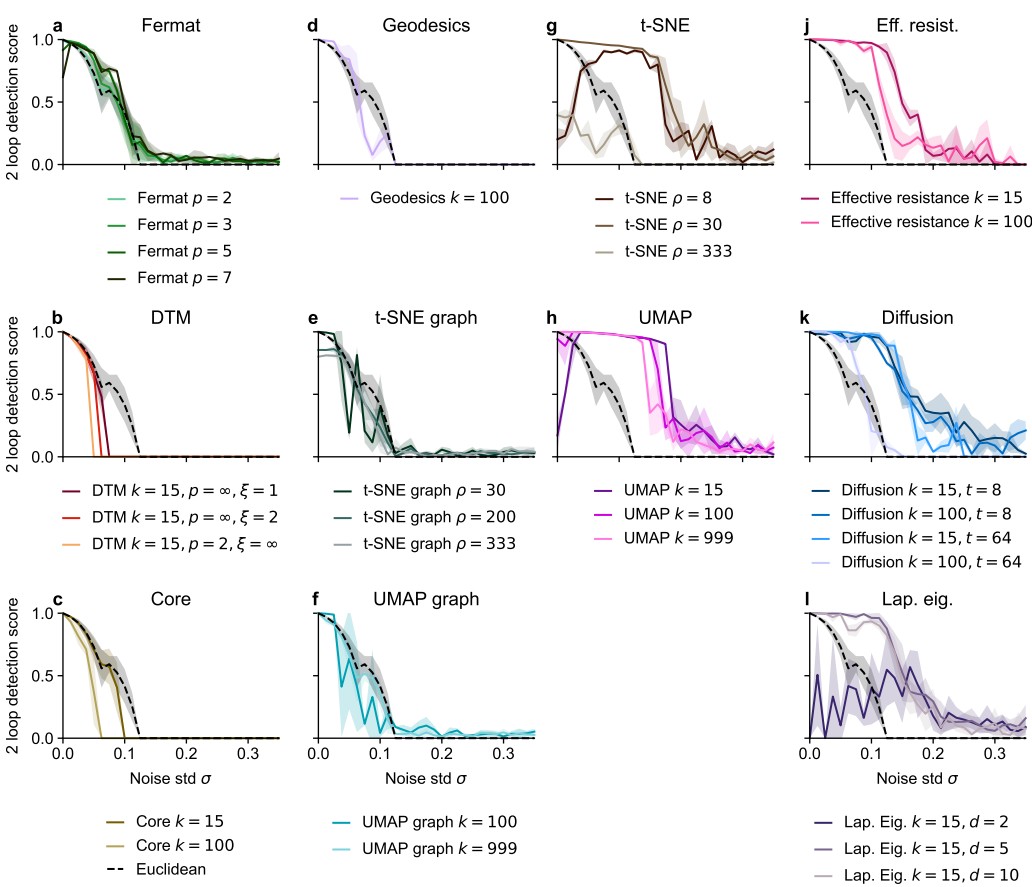

Figure S14: 2 loop detection score for persistent homology with various distances on two interlinked circles in ambient $\mathbb{R}^{50}$. Spectral and embedding methods performed best, but the latter sometimes had issues in the low noise setting.

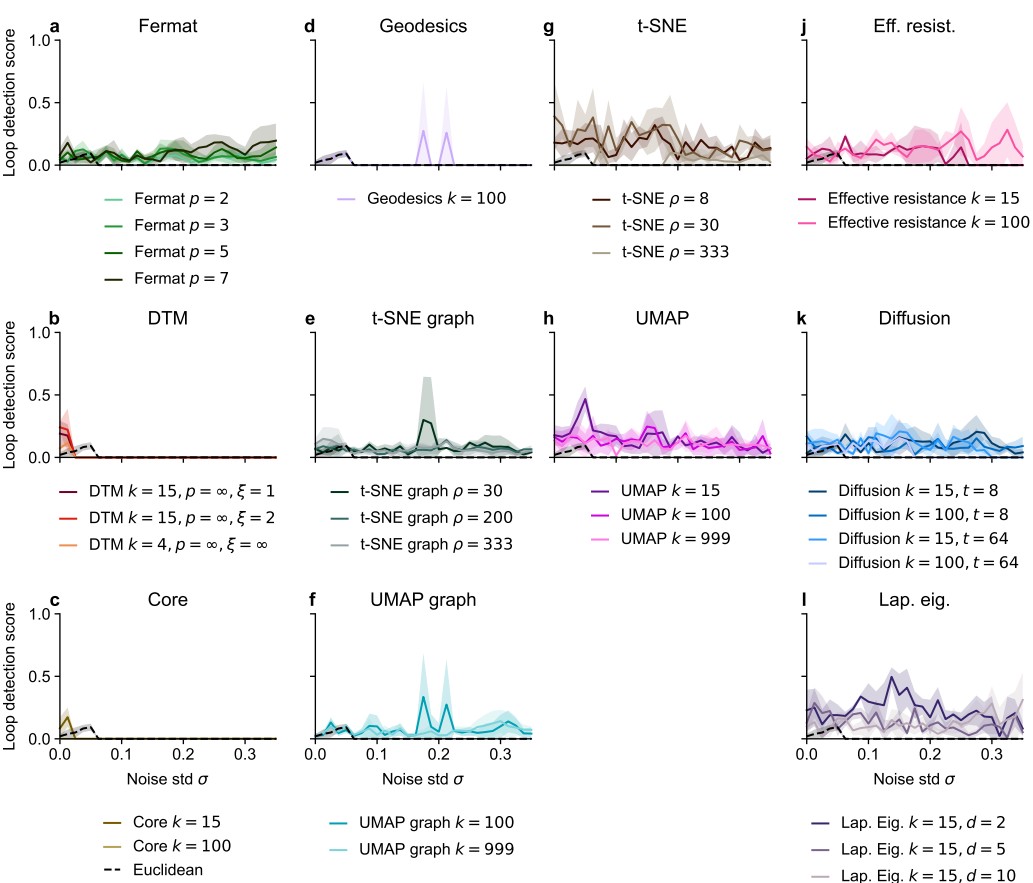

Figure S15: Loop detection score for persistent homology with various distances on a noisy sphere in ambient $\mathbb{R}^{50}$. Most methods pass this negative control.

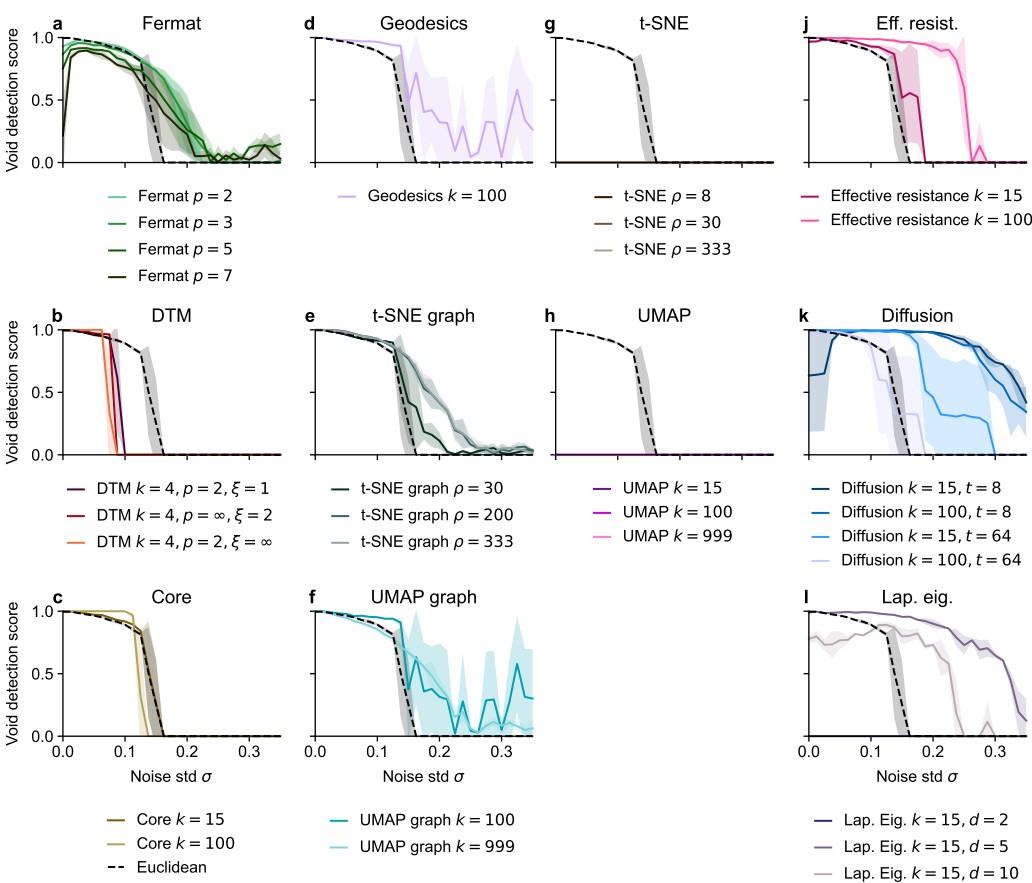

Figure S16: Void detection score for persistent homology with various distances on a noisy sphere in ambient $\mathbb{R}^{50}$. Methods relying on 2D embeddings did not find the loop for any noise level. Spectral methods performed best.

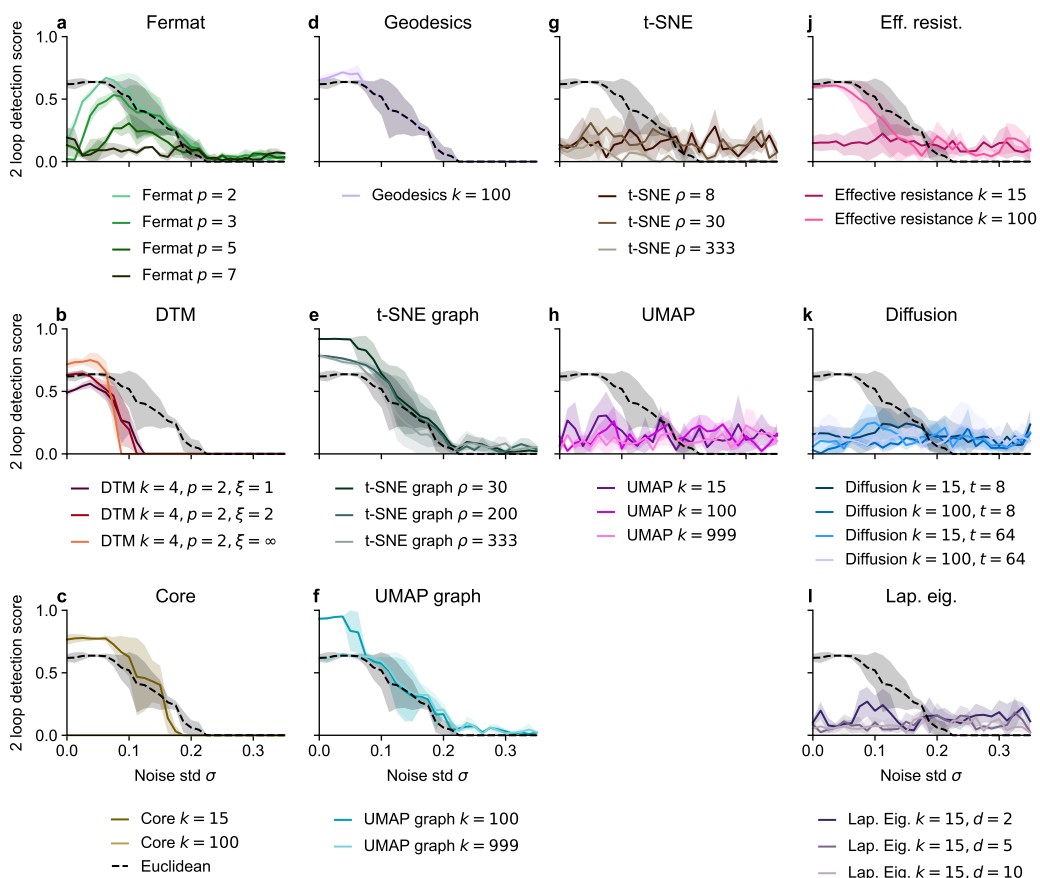

Figure S17: 2 loop detection score for persistent homology with various distances on a noisy torus in ambient $\mathbb{R}^{50}$. All methods struggled here, and only DTM, $t$-SNE graph and UMAP graph improved noticeably over the Euclidean distance. On a denser sampled torus effective resistance and diffusion distance outperformed other methods (Fig. S18.

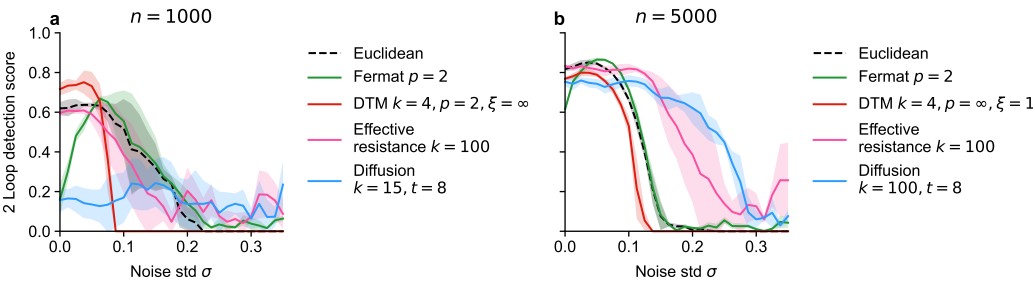

Figure S18: 2 loop detection score for persistent homology with various distances on a noisy torus of different number of points $n$. For more points, all methods perform better as the shape of the torus gets sampled more densely. The difference in performance is particularly striking for the spectral methods which outperform the others for $n = 5\,000$ points, but did not for $n = 1\,000$.

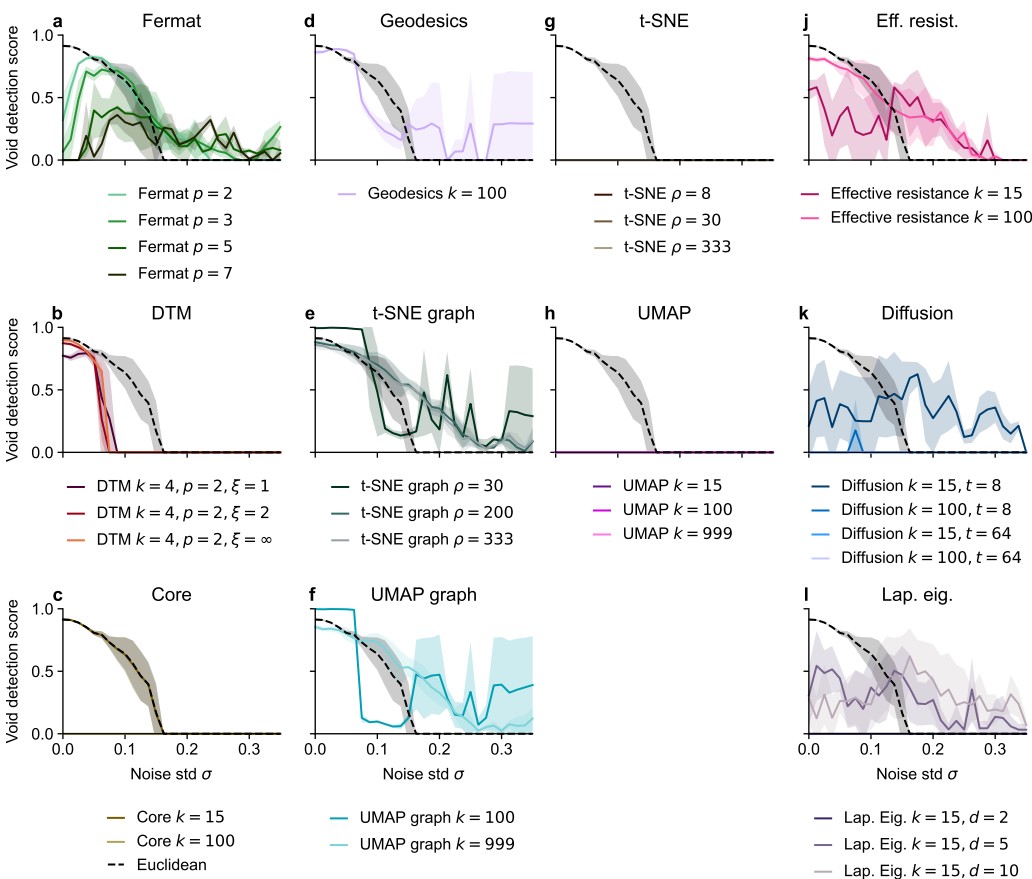

Figure S19: Void detection score for persistent homology with various distances on a noisy torus in ambient $\mathbb{R}^{50}$. Methods relying on 2D embeddings do no find the void for any noise level. Only $t$-SNE graph and UMAP graph could reliably improve above the Euclidean distance and only for low noise levels. However, they had unstable behavior for higher noise levels, resulting in high uncertainties. We suspect that a higher sampling density would benefit effective resistance and diffusion distance a lot, but the computational complexity of persistent homology makes such experiments difficult.

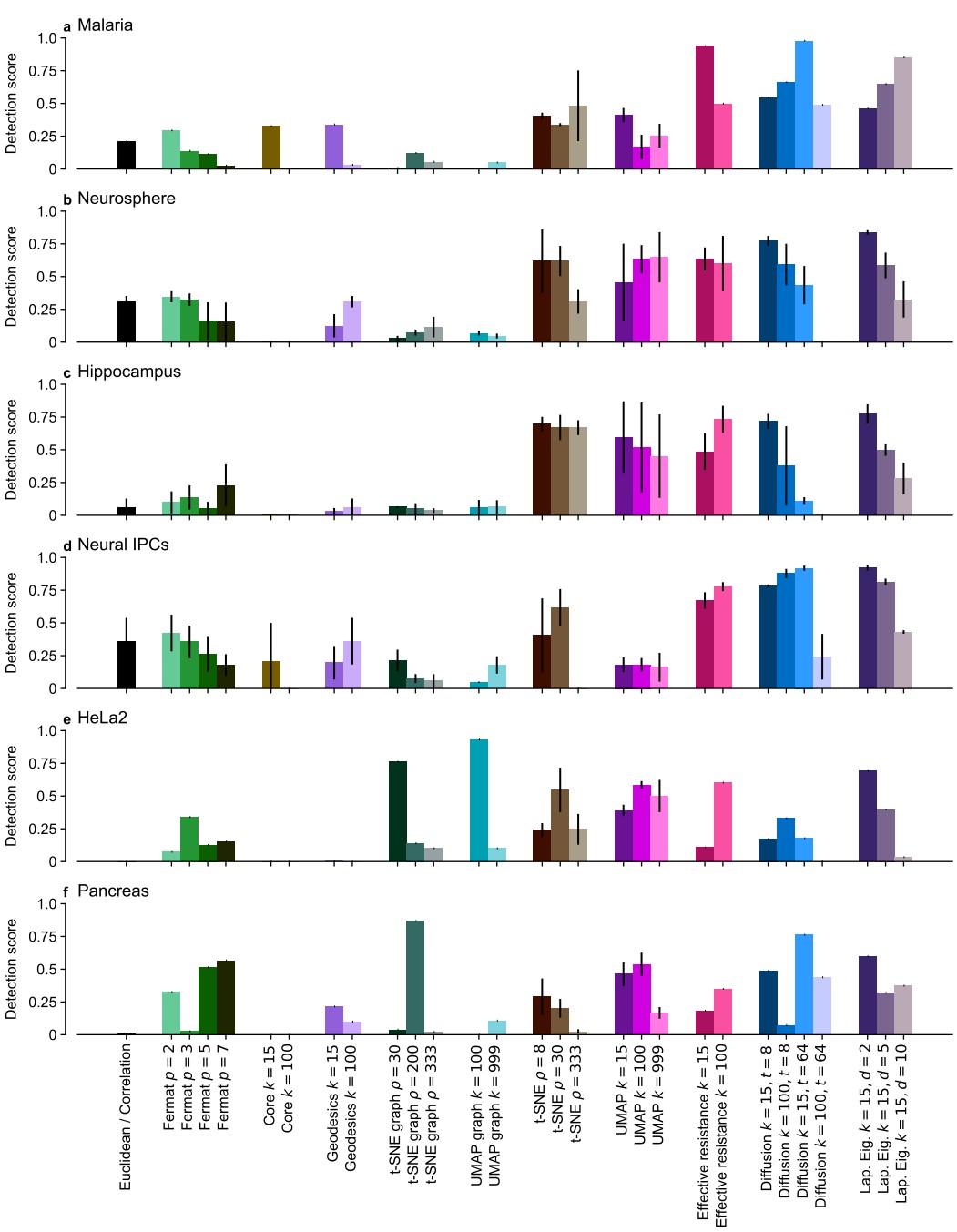

Figure S20: Detection scores for all hyperparameter settings for all six single-cell datasets. We omitted DTM as no setting passed the thresholding on any dataset. The black bar refers to correlation distance on the Malaria dataset and to Euclidean distance on the others. Extension of Figure 9. $t$-SNE graph and UMAP graph can perform very well, but are very hyperparameter dependent. Their embedding variants often perform well, but collapse on some datasets. The spectral behave similarly, but perform better on average.

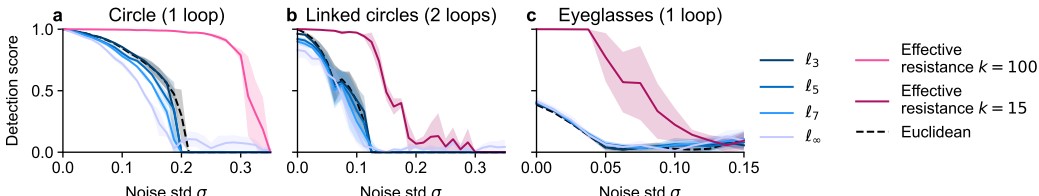

Figure S21: Detection scores for effective resistance, Euclidean distance, and various $\ell_p$ distances on the noisy circle, interlinked circles, and the eyeglasses dataset in ambient $\mathbb{R}^{50}$. For $p > 2$ the $\ell_p$ distance becomes non-isotropic which outweighs any positive effect of accumulating less of the noise. The detection scores are on par or worse than those of the Euclidean distance.

