# OpenReview forum: "Persistent homology for high-dimensional data based on spectral methods"
_ICLR.cc/2024/Conference — Submitted to ICLR 2024_

### Official Review · Reviewer_BdHd · 2023-10-29

**Soundness:** 3 good
**Presentation:** 3 good
**Contribution:** 2 fair
**Rating:** 6
**Confidence:** 5

**Summary:**

The paper addresses an important challenge: the efficient adaptation of persistent homology tools to point clouds in high-dimensional spaces. Many crucial datasets across different domains are presented in this format, but the curse of dimensionality poses a substantial obstacle to the application of an effective feature extraction method, namely persistent homology, in this context. The authors introduce two novel distance measures to facilitate the efficient utilization of persistent homology in this scenario. They validate their approach with several experiments.

**Strengths:**

1. The authors address a significant problem in the ML domain related to high-dimensional point clouds. While there exist various dimension reduction techniques for handling such data, a potentially effective feature extraction method, persistent homology, cannot be facilitated efficiently in the presence of noise in high-dimensional data, as they lead to substantial impacts on the output. To tackle this issue, the authors introduce a novel approach, leveraging two distinct distance measures, effectively circumventing the challenges posed by noise.

2. The authors conduct a comprehensive analysis by performing multiple experiments to validate their approach. They establish the effectiveness of their method by comparing it with several alternative approaches across various datasets.

**Weaknesses:**

1. The main concern with the proposed method is combining kNN graphs with persistent homology. While this provides a solution to deal with noisy data, it unfortunately introduces a more subtle problem: The outliers. Employing kNN in such datasets, immediately makes outliers a big problem for the persistent homology, as if one uses kNN graph distance (even with proposed modifications), any outlier will bring extra unnecessary topological features with high persistence up to dimension k-1.

This is one of the main obstacles to employing PH in high dimensions, and to tackle both outlier and noise problems, many researchers in the field try to employ multiparameter persistence methods.

2. The "hole detection score" performance metric proves to be a valuable measure when dealing with datasets featuring a single significant topological feature. However, its effectiveness diminishes when there are multiple topological features of similar sizes in the data. Hence, this metric becomes most useful when one already has prior knowledge of the dataset's topological structure, as illustrated in Figures 4 and 5. On the other hand, if the hidden topological structures within the data remain unknown, the performance metric loses its significance, as the suppression of similar-sized features (resulting in low detection scores) could be a desirable outcome for that specific dataset. Therefore, it's essential to approach the results presented in Figure 9 with caution, as they may potentially be misleading.

3. I greatly appreciate the diverse experiments conducted across various settings with different distance measures. However, what I'm particularly eager to observe is the application of your approach to a meaningful classification problem within high-dimensional point clouds. e.g., analyzing its performance in distinguishing between cancer and normal tissues in single-cell RNA sequencing datasets. Such an experiment would provide a more concrete measure of effectiveness. It would be beneficial to assess the performance of persistent homology vectorizations with various PH settings, such as Euclidean, UMAP, and your proposed distances, in this context. This approach would offer a more robust performance evaluation and further validate the effectiveness of your methodology.

4. The paper's organization could be enhanced. I suggest gathering Sections 3, 4, and 5 as subsections under the "Background" section. Placing Section 7 before the "Experiments" section should enhance the flow. Additionally, collecting Sections 6 and 8 into an "Experiments" section might improve clarity. However, it's important to note that this suggestion is a matter of preference and should be considered a minor comment, subject to your discretion.

**Questions:**

See Weaknesses.

---

> ### Author Response · Authors · 2023-11-17
> **Thank you for the constructive review!**
>
> Dear reviewer BdHd,
>
> thank you for you detailed review. We are happy that you found the problem setting we tackle "significant", our approach "novel", and our analysis "comprehensive". We will address your concerns in the following.
>
> **Outlier sensitivity:**
> This is an interesting question, but we disagree about outliers being a problem for kNN-graph-related methods. First, all methods but Euclidean distance and Fermat distances, in particular DTM-based filtrations which are designed to handle outliers, rely on the kNN graph in some way. However, they do not use the raw distances as edge weights, but aggregate these in a way that avoids sensitivity to outliers. As an illustrative example, consider effective resistance and the 2-NN graph. In case of single outlier $o$, it would have its nearest neighbors $p_1$ and $p_2$ on the main data structure, say the noisy circle. These neighbors might be far away in the Euclidean distance (which is why Euclidean distance can suffer from outliers). But the 2-NN graph on the circle is densely connected, such that the effective resistance distance between $p_1$ and $p_2$ is small. Thus, the triangle $op_1p_2$ is very slim and does not lead to a highly persistent 1D hole.
>
> To validate this, we now conducted additional experiments with uniformly distributed outliers in the bounding box around the noised circle (Appendix G). In low ambient dimension, outliers hurt the Euclidean distance at low noise levels a lot. Effective resistance remained much more robust against outliers.
>
> Second, as we describe in Section 4, outliers cease to be a big problem in high-dimensional spaces, as they are necessarily extremely sparse. It is highly unlikely that an outlier point falls, say, into the middle of circle in high ambient dimensionality. We backed this up by additional experiments, showing that even the Euclidean distance is robust to adding up to 10% uniformly distributed outliers in $\mathbb R^{50}$. Effective resistance continues to outperform the Euclidean distance even in the presence of outliers in this setting.
>
>
> **Usefulness of our detection score:**
> We completely agree that our detection score is only useful if ground truth on the number of holes in the data is available. This is akin to most metrics in machine learning, which only work in reference to a ground truth. If ground truth is available, our metric works for any number of holes in the data as it targets the gap between the $m$-th and $(m+1)$-st highest persistence for $m$ ground truth holes. For the synthetic datasets ground truth is known by construction. For the single-cell datasets, we made an effort to only benchmark datasets for which the number of loops is clearly determined by the biology of the sample. For the malaria dataset there are two cycles: the parasite reproduction cycle in red blood cells and the parasite transmission cycle between humans and mosquitos as described in Section 7.2. The other five datasets are created so that the gene expression variation comes from the cell division cycle (mytosis), that is, the process of a cell dividing into two copies of itself. Therefore, we expect one loop in these datasets.
>
> **Downstream single-cell experiment:**
> We are happy that your appreciate our diverse set of experiments! An experiment like the one you suggested is indeed a natural next step. Thanks for making this suggestion. However, this type of experiments is future work and could not be conducted during the rebuttal period (and would probably be of more interest for biologists, as opposed to ML researchers). In addition to using an effective form of persistent homology in a classification pipeline, we also believe that looking for holes in complex single-cell datasets would be an interesting and important research direction. But in order to trust persistent homology as a tool for exploring the structure of single-cell data, we first had to validate which approach to PH is trustworthy. This is the aim of our paper.
>
> **Restructuring suggestions:**
> Thank you for these constructive suggestions! The section on the curse of dimensionality for persistent homology is a new contribution and not background. Since the flow from persistent homology (Section 3) to its curse of dimensionatlity (Section 4) and finally the resolution with spectral methods (Section 5) seems fitting, we kept these chapters as they were. We did, however, move the more theoretical section on the relation between spectral distances up and combined the two experimental sections, as you suggested.
>
> If you have are any further concerns, please do let us know and we will be happy to answer them!

---

> > ### Comment · Reviewer_BdHd · 2023-11-20
> >
> > Thanks for your response. I greatly appreciate your clarifications in other questions, but I disagree with your response to the outlier problem. Yes, in your toy example, the triangle will be very slim, but it will still have high persistence as some edges are long. In any case, I think the outlier problem can be resolved with additional pre-processing steps. So, I am increasing my score. Good luck with your submission.

---

> > > ### Author Response · Authors · 2023-11-21
> > >
> > > Thank you very much for getting back to us! We are happy that you found our reply helpful and greatly appreciate you raising your score.
> > > We updated the section on outlier sensitivity (Appendix G) and added more methods. Still, outliers are only a small problem in high dimensions, such that the spectral methods maintain their good performance. In low dimension the kNN-graph-based method DTM excels at handling outliers, but the spectral methods still perform well. If you have any questions or comments on these experiments, please let us know!

---

### Official Review · Reviewer_qoEF · 2023-10-31

**Soundness:** 3 good
**Presentation:** 3 good
**Contribution:** 2 fair
**Rating:** 3
**Confidence:** 4

**Summary:**

In this paper, the authors investigate the problem of applying persistent homology (PH) to high dimensional noisy data. They argue that in high dimension, Euclidean distance is not suitable for creating filtrations involved in computing PH since pairwise distances tend to be similar for noise generated from Gaussians. Instead, the authors propose to use knn graphs as well as their corresponding (modified version of) effective resistance or spectral diffusion distance to induce filtrations. They empirically tested their idea on a toy synthetic dataset as well as 6 single-cell RNA-sequencing datasets.

**Strengths:**

- The clarification of how the curse of dimensionality affects the computation of PH is very clear and easy to understand. This clear explanation can help guide future studies on using PH with high-dimensional data.
- The empirical comparison between the effective resistance and the corrected effective resistance is interesting. It hints that the corrected version might be the better choice for real-world uses.

**Weaknesses:**

- Although the effect of dimensionality is shown empirically, there is a lack of theoretical results in explaining the curse of dimensionality to PH. See the question section for more details.

- While the paper provides insightful observations, the novelty aspect seems to be limited for the expectations of an ICLR publication.
  - The major approach is to first use spectral methods for dimension reduction implicitly (as only the distances instead of the coordinates are used for PH). This is quite natural and has been studied intensively in manifold learning. I don't think applying persistent homology to data after this type of dimension reduction is novel enough for publication in ICLR.
  - The formula in Proposition 1 is nice to have. However, as pointed out by the authors themselves, this is simply a clarification of the existing claim in von Luxburg et al. (2010a) that the corrected effective resistance is a squared Euclidean distance.

**Questions:**

In the curse of dimensionality part, I wonder if the authors can provide some theoretical results to support their claim. For example, can they show that the length of 1-dim barcodes is bounded by some function of the dimension with high probability for the circle data with Gaussian noise?

---

> ### Author Response · Authors · 2023-11-17
> **Rebuttal by the authors**
>
> Dear reviewer qoEF,
>
> thank you for writing your review! We appreciate that you found our description of the curse of dimensionality for persistent homology "very clear and easy to understand". We address your concerns below:
>
> **Novelty**:
> Please confer the general comment on novelty. Briefly put, we believe that solving a new, relevant problem with a novel combination of existing methods can be a more valuable contribution than inventing a new method specific to the problem. Please note that the corrected effective resistance had NEVER been combined with persistent homology before (as we now explicitly state in the related work section). Also, all these methods have never been applied to high-dimensional data with high-dimensional Gaussian noise, which is a setting very important in many practical applications. Finally, our theoretical analysis of corrected effective resistance and its relationship to diffusion distances is also novel.
>
> **Explicit formula for corrected effective resistance:**
> Our explicit formula enabled us to relate corrected effective resistance to diffusion distances. Using it, we found, for instance, that the corrected effective resistance decays eigenvectors with high eigenvalues more aggressively than the uncorrected version, but less aggressively than diffusion distances. We consider it an important theoretical result of our paper.
>
> **Theoretical treatment of curse of dimensionality:**
> This is a very interesting, but difficult question, and is beyond the scope of this rebuttal. Our Section 4 provides a qualitative treatment of this curse of dimensionality (which we have not seen explicitly discussed in the literature before). A quantitative treatment remains for future work. In fact, motivated by your question, we started working on it and have some promising ideas for a theoretical treatment, but would need more time to work on it than the rebuttal period allows.
>
> If you have are any further concerns, we will be happy to answer them!

---

> > ### Comment · Reviewer_qoEF · 2023-11-22
> >
> > I thank the authors for addressing my comments. However, I stand by my point that the current version of the paper is lacking novelty.
> >
> > To summarize, the paper posits that (1) persistent homology is affected by curse of dimension, (2) spectral methods could be applied to reduce dimension (the authors didn't present in this way but this is basically what the spectral / effective resistance methods are doing), (3) apply persistent homology afterwards. While the introduction of formula of corrected effective resistance in section (2) is interesting, I believe that a more substantial contribution is required in (1) to elevate the novelty of the paper. While I am pleased that the authors have considered enhancing section (1) per my previous suggestion, as it stands, I decide not to change my score.

---

> > > ### Author Response · Authors · 2023-11-23
> > >
> > > Thank you for getting back to us and summarizing your position! Naturally we are disappointed but we appreciate your feedback.
> > >
> > > We would only like to comment on one aspect, to prevent misunderstandings:
> > >
> > > **Effective resistance and diffusion distances are not doing dimensionality reduction:**
> > >
> > > We would like to stress that effective resistance and diffusion distance, unlike the spectral method Laplacian Eigenmaps, do *not* need an embedding dimension as input. This is an important advantage as it reduces the number of tunable hyperparameters. Both effective resistance and diffusion distance can only be realized exactly by an $(n-1)$-dimensional embedding.
> > >
> > > We agree that in some sense, they could be seen as a `soft' version of dimensionality reduction, with some dimensions only contributing little to the spread of the data --- but the "effective embedding dimension" (the number of dimensions decayed only a little, see Section 5) is determined in a data-dependent way.
> > >
> > > In particular, if the data is truly high-dimensional, this will be reflected in the effective resistance / diffusion distance. For instance, consider the 100-NN graph of $n=1\,000$ points sampled from an isotropic $50$-dimensional normal distribution. Plotting the decay curves (like in Section 5), we saw a clear elbow at $d=50$ for both effective resistance and diffusion distance with $t=8$ (not shown in the paper), meaning that both methods would not actually reduce the dimensionality for this truly high-dimensional dataset. Similarly, if the $k$-NN graph of a dataset was (close to) the complete graph (e.g., due to large $k$ or very high ambient dimension), both diffusion distance and effective resistance would correspond to a truly $(n-1)$ dimensional embedding as all but the first eigenvalue of the graph Laplacian would be the same.

---

### Official Review · Reviewer_k35W · 2023-10-31

**Soundness:** 2 fair
**Presentation:** 2 fair
**Contribution:** 3 good
**Rating:** 3
**Confidence:** 4

**Summary:**

The article presents a new approach to compute persistent homology of given set of data points. The approach builds the simplicial complexes using spectral distances, such as diffusion distance and effective resistance, on the k-nearest-neighbor graph of the data. The paper suggests persistent homology computed based on spectral distances correctly detect the topology of the data even in the presence of high-dimensional noise. A closed form formula is also derived for the effective resistance (distance) based on the eigendecomposition of the kNN graph Laplacian. Numerical results are presented on different synthetic and single cell RNA-sequencing datasets to illustrate the performance of the proposed method.

**Strengths:**

Strengths:
1. The paper demonstrates that spectral distances such as diffusion distance and effective resistance perform well in detecting cycles in high-dimensional noisy data.
2. The eigendecomposition based effective resistance formula is interesting and might be of independent interest.
3. The paper presents several interesting numerical experimental results.

**Weaknesses:**

Weakness:
1. The presentation can be improved. The paper might be hard to follow for non experts.
2. The main methodology proposed is not well-defined.
3. The novelty and advantages of the proposed method are not clear.

**Questions:**

The paper studies an interesting problem of cycle detection in high dimensional noisy data. The findings related to the use of different distance metrics is interesting.

I have the following comments:

1. The presentation can be improved. Currently, the main methodology and several aspects are not at all clear.

First, it appears the loops and cycles are detected using a detection score that depends on what is termed as m-th most persistent features . But it is not clear what does persistences p_m of the m-th most persistent features mean? How are these calculated? How are these persistent features related to the loops/holes?  Given a distance metric, how are these features and the detection score computed? If the underlying graph structure for the input data points are not given, how is the k-NN graph constructed? These details are missing.


Next, typically, in persistent homology (as described in the intro), the resolution (radius of the ball around the datapoints) is increased, and the Betti number or other homology related features are computed. However, in this paper, it is not clear what exactly is computed. How are the holes/loops detected and what is the persistence (birth-death of holes) with respect to. Is the resolution scale with respect to the different distance metrics considered? If so, what is the role of the K-NN graph? The graph connection is predefined to find these distances in this case.

2. In the related works section, many previous works have been mentioned where similar distance metrics have been used for persistence
Homology. How does the proposed method differ from them is not clearly described.

3. The advantage of the proposed method is also not clear. It appears some of the recent dimensionality reduction methods such as t-SNE and UMAP seem to perform better at detection holes than the proposed method and these method should also have lower computational cost.

4. In the datasets, the dimension of the holes detected is not clear. Note that a torus has 2 2D holes and 1 3D hole. High dimension holes are formed by higher order simplices (a k-dimensional hole has (k-1)-simplices as its boundary). Here cycles/holes only seem to consider edges, and not higher order simplices. Is this correct?
How is the Vetoris-Rips complex constructed? Given n points, the complex can have large number of higher order simplices, and detecting high dimensional holes is very expensive (can be exponential cost and is an NP hard problem).
Again, it is hard to understand due lack of details.

Overall, the merits of the paper is difficult to figure out.

---

> ### Author Response · Authors · 2023-11-17
> **Rebuttal by the authors (1/n)**
>
> Dear reviewer k35W,
> thank you for reading our paper and writing the review! Please note many of the details you asked about are all present in the manuscript. Our overall approach is conceptionally standard and close to previous works such as Bendrich et al. 2011, Anai et al. 2020, Fermandez et al. 2023: We compute a pairwise distance matrix, build the filtered Vietoris-Rips complex based on that distance matrix, and compute persistent homology of this filtered simplicial complex. Computationally, this was done using the `ripser` library. We only needed to compute various distances matrices and pass them to `ripser`. We have described this approach in detail in Section 3.
>
> That said, we will make an effort to resolve any confusion in our replies below!
>
> >But it is not clear what does persistences p_m of the m-th most persistent features mean?
>
> We describe in Section 3 that the persistence of a feature is the difference between its death and birth time: "Each hole has associated birth and death times $(\tau_b, \tau_d)$, i.e., the first and last filtration value $\tau$ at which that hole exists. Their difference $p = \tau_d - \tau_b$ is called the *persistence* or *life time* of the hole and quantifies its prominence."
>
> The m-th most persistent feature is that with the m-th highest persistence.
>
>
> >How are these persistent features related to the loops/holes?
>
> We describe persistent homology in Section 3. A $\delta$-dimensional feature that persists from $\tau_b$ to $\tau_d$ corresponds to $\delta$-dimensional hole in the simplicial complex between filtration values $\tau_b$ and $\tau_d$ which in turn can be thought of $\delta$-dimensional holes that persist in a union of balls around the data points of radius $\tau_b$ to $\tau_d$. We stress that we explain this in Sec.3.
>
>
> >Given a distance metric, how are these features and the detection score computed?
>
> We use the package `ripser` to compute the persistence diagram, as stated in Section 3 and Appendix F. This is a standard package for the computation of persistent homology. Given a pairwise distance matrix, ripser computes the persistent homology of the associated Vietoris-Rips complex. The output, a persistence diagram, contains the information about detected holes / loops. When we talk of the persistence of a feature, say, for effective resistance, we mean a feature that was computed by the ripser package when providing it with the effective resistance distance as input.
> We describe the computation of our detection score in Section 7 (paragraph "Performance metric"): If the ground truth number of holes of dimension $\delta$ is $m$, we take the persistences $p_m$ and $p_{m+1}$ of the $m$-th and the $(m+1)$-st most persistent feature in dimension $\delta$ and compute the ratio $s_m=(p_m -p_{m+1}) / p_m$ as our detection score. In particular, all these details of our approach are present in the manuscript.

---

> ### Author Response · Authors · 2023-11-17
> **Rebuttal by the authors (2/n)**
>
> > How are the holes/loops detected and what is the persistence (birth-death of holes) with respect to. Is the resolution scale with respect to the different distance metrics considered?
>
> Holes are detected with the ripser package (Section 3). Indeed, the resolution scale is with respect to the different distance metrics considered. This is explained in Section 3, where we write that the ball-growing procedure is made computationally tractable by the use of a filtered simplicial complex, that we only work with the Vietoris-Rips complex, and that for this complex it suffices to specify a pairwise distance metric. Later in Section 7, paragraph "Distance measures" we write that we consider twelve distances as input to persistent homology.
>
>
> >  If so, what is the role of the K-NN graph?
>
> All distances other than Euclidean and Fermat depend on the kNN graph in various ways. We described the all distances briefly in Section 7 and in detail in Appendix C. Note that the input is always a point cloud, never a predefined graph. If a distance relies on the kNN graph, we compute it from the point cloud (usually with a parallelized brute-force approach implemented with the package `pykeops`). The exact role of the kNN graph depends on the distance. For instance, the distance we call Geodesics is the shortest path distance on the kNN graph.
>
> > The graph connection is predefined to find these distances in this case.
>
> We do not quite understand what you mean by this sentence. Could you please elaborate?
>
>
> > How does the proposed method differ from them is not clearly described.
>
> Overall, our paper is about evaluating which (possibly existing) methods perform well for detecting holes in the presence of high-dimensional noise, a problem setting that has not been addressed as we state in the first paragraph of the related work section. We added a sentence to the related work section clarifying that persistent homology had so far never been combined with the corrected effective resistance of von Luxburg et al. 2010a, one of the methods that we find performed best in our problem setting.
>
>
> > t-SNE and UMAP seem to perform better at detection holes and these method should also have lower computational cost.
>
> Please see the general comment on the performance of t-SNE and UMAP.  t-SNE and UMAP are typically *slower* that effective resistance or diffusion distance. Both first compute the kNN graph of the data, but t-SNE and UMAP then perform an iterative optimization scheme, while effective resistance only needs to compute the pseudoinverse of a matrix and diffusion distances only need to power a matrix.
>
> Furthermore, t-SNE and UMAP are typically used for 2D embeddings, and it is not possible to detect higher-order homologies (e.g. H2 homologies such as voids) using a 2D embedding.
>
>
> > Here cycles/holes only seem to consider edges, and not higher order simplices. Is this correct?
>
> No, this is not correct. We always consider simplices of the dimension needed to compute the type of holes (loops, voids) that we are interested in. For instance, if we compute 2D holes (voids), we consider simplices of dimensions 1, 2, and 3. This is handled by the ripser package.
>
> > How is the Vetoris-Rips complex constructed?
>
> The Vietoris-Rips complex is a standard construction in topology. We describe it in Section 3: "... the Vietoris–Rips complex, which includes an n-simplex (v0, v1, ... , vn) at filtration time $\tau$ if the distance between all pairs vi, vj is at most $\tau$. Therefore, to build a Vietoris–Rips complex, it suffices to provide pairwise distances between all pairs of points."
>
> In particular, the Vietoris-Rips complex on $N$ points can contain simplices of dimension up to $N-1$. Whether a particular simplex, even a high-dimensional one, is contained at time $\tau$ only depends on pairwise distances. The construction of the Vietoris-Rips complex is handled by the `risper` package. One of its arguments is the maximal feature dimension that should be computed. Based on this, it only creates those simplices necessary, so that in practice very high-dimensional simplices are not constructed.
>
> > Given n points, the complex can have large number of higher order simplices, and detecting high dimensional holes is very expensive.
>
> As mentioned above, this part is abstracted away by the ripser package. However, since we always set a maximal dimension for the topological features that should be computed (2 for void detection and 1 for loop detection), no simplices of dimension 4 or higher are needed. Indeed, we observe that then we compute persistent homology of 2D holes (voids) the run time increases a lot. We discussed this in Section 8: "One limitation of persistent homology is its computational complexity. ...."
> We also reported exemplary run times in Table S3, again showing that void detection is much more costly than loop detection.

---

> > ### Author Response · Authors · 2023-11-17
> > **Rebuttal by the authors (3/n)**
> >
> > > The presentation can be improved. The paper might be hard to follow for non experts.
> >
> > Following the recommendation of reviewer BdHd, we have grouped the two sets of experiments into one section, improving the flow of the paper.
> >
> > If there are any remaining concerns, we would be happy to answer them!

---

> ### Comment · Reviewer_k35W · 2023-11-22
>
> I thank the authors for their thorough responses to all reviewers' comments. These are very helpful, and make many of the details related to the methodology now more clear.
>
> However, I think the current version of the paper falls short due to the following reasons, and I encourage the authors to consider these points when revising the draft:
>
> (a) It appears the readers of the paper are expected to know what the ripser library package does, and should be familiar with the "standard techniques" presented in the existing literature listed in this response to understand and follow the paper. Many of the details in the responses to the reviewers are clearly missing.
>
> (b) In the response above, authors say "We have described this approach in detail in Section 3."
> However, I do not see many of the details that are discussed here in the response, in section 3 (which is half a page) of the paper. In the section, there is just a single sentence, "We compute persistent homology via the ripser package", and the readers are expected to know what this package does.
>
> (c) Also, in section 3 and the response above, it is suggested, "we use the Vietoris–Rips complex, which includes an n-simplex ..." As noted by the authors above, given $N$ points, a Vietoris–Rips complex can include simplices of order up to $N-1$. However, it appears the full Vietoris–Rips complex (with simplices of all orders) is never constructed, and only a simplicial complex with simplices of order up to $k=2$ or $3$ is considered. This makes sense (due to the cost), but was never clear in the paper.
> Similarly, the paper just says that persistent homology involves computing the homology groups of the union of all balls in order
> to find the holes. The dimension of these holes are not specified  (mention of $\delta$-dimensional feature and holes only appear here in the response). Note that the general definition of  persistent homology is constructing the full simplicial complex and finding/counting holes of all dimensions that persist across different scales $t$. The authors' response above suggests in the paper $\delta =1$ or $2$ are considered.  Again, this is perfectly fine (since higher order holes are expensive to estimate), but is never discussed in the paper.
>
> Since all these details are currently missing in the paper, a reader would just assume the full complex and holes of all dimensions are computed. Since constructing the full Vietoris–Rips complex and computing all Betti numbers will be very expensive (could be of exponential cost), a natural conclusion is to assume that the overall cost of the proposed method will be extremely high.
>
> (d) The role of k-NN graph as related to persistent feature computation is still not clear to me, and section 7 does not contain much details related to this as suggested in the authors' response above.
>
> (e) In the response above, authors say "our paper is about evaluating which (possibly existing) methods perform well for detecting holes in the presence of high-dimensional noise". This should have been the main premise of the presentation, but in the abstract and introduction, this fact is never made clear. Therefore, naturally reviewers have asked about novelty and relation to existing methods.
>
> Overall, the current version of the paper is very difficult to follow and the main contributions, methodology, and significance are all not really clear, as also suggested by other reviewers' comments. Hence, I am keeping my current score.

---

> > ### Author Response · Authors · 2023-11-22
> >
> > We appreciate your response! We now made some further clarifications in the manuscript, but are also puzzled by some of your remaining criticisms.
> >
> > > the general definition of persistent homology is constructing the full simplicial complex and finding/counting holes of all dimensions that persist across different scales $t$. The response below suggests in the paper $\delta =1$ or $2$ are considered. Again, this is perfectly fine (since higher order holes are expensive to compute), but is never discussed in the paper.
> >
> > This is not true. We have clearly stated that we only consider 1D and 2D holes (loops and voids), in multiple places in the paper. Section 2: "Here we only investigate the detection of higher-dimensional (1D and 2D) holes with persistent homology". Section 7.1.1: "For each resulting dataset, we computed persistent homology for loops and, for the sphere and the torus, also for voids."
> >
> > That said, to make this even clearer, we now added the sentence "We never computed holes of dimension $3$ or higher." at the end of section 7.1.1. We also rephrased the potentially confusing sentence "The output of persistent homology is a set of holes for each dimension" to "Persistent homology consists of a set of holes for each dimension. We limit ourselves to loops and voids."
> >
> > We hope this resolved this confusion!
> >
> > > It appears the readers of the paper are expected to know what the ripser library package does
> >
> > Relying on a computational package (such as ripser, PHAT, GUDHI, etc.) for the computation of persistent homology is completely standard. Many papers such as Fernandez et al. (2023), Turkes et al. (2023), Anai et al. (2020) also only mention the package used without giving any further details. While these packages might differ in speed and implementation details, the resulting persistent homology only depends on the filtered simplicial complex. We describe which simplical complex we use and explain on an intuitive (but exact) level what persistent homology is (holes in a union of growing balls; Figure 2). Omitting the exact details of how ripser implements the computation of persistent homology is in the best interest of our readers. Of course, we reference the ripser paper (20 pages in itself) for the interested readers.
> >
> >
> > > The role of k-NN graph as related to persistent feature computation is still not clear to me, and section 7 does not contain much details related to this as suggested below in the response.
> >
> > We are frankly puzzled by this criticism. The input to the persistent homology pipeline is a distance matrix and we explore several different distances. Some of them rely on the $k$NN graph, e.g., Geodesics are the shortest path distance on the kNN graph. We describe all distances in Section 7 and we mention "$k$-nearest neighbor" or "$k$NN graph" in four sentences in that paragraph. Moreover, we spend more than two pages discussing all the distances in detail in Appendix C.
> >
> > > In the response below, authors say "our paper is about evaluating which (possibly existing) methods perform well for detecting holes in the presence of high-dimensional noise". This should have been the main premise of the presentation, but in the abstract and introduction, this fact is never made clear.
> >
> > This criticism surprises us as we do mention this fact both in the abstract and the introduction:
> >
> > In the abstract we write "vanilla persistent homology becomes very sensitive to  noise [...] The same holds true for most existing refinements. [...] we find that spectral distances allow persistent homology to detect the correct topology even in the presence of high-dimensional noise".
> >
> > The introduction ends with the list of contributions one of which is "a synthetic benchmark, with spectral distances outperforming state-of-the-art alternatives". We also mentioned our "synthetic benchmark" in the discussion.

---

> ### Comment · Reviewer_k35W · 2023-11-22
>
> I thank the authors for the response and the changes made to the paper.
>
> But, I think the authors are missing the main point reviewers are trying to make, that the current paper presentation does not clearly convey different aspects (e.g., main contributions/premise, methodology used, and significance) of the presented work.
>
> One of the key duties of reviewers is to provide suggestions to help improve the paper.
>
> Mentioning something in some part of the paper, is not equivalent to a clear explanation or discussion, right?
> Readers should not be excepted to guess, or go back and forth/search for basic key details. For example, as the authors above say, the fact that only 1D and 2D holes are computed, is just mentioned (and one had to search for it) as a pass-by sentence, but in the main methodology section, this was not discussed.
> Same with the k-NN graph, indeed it is mentioned at several places, but the nice discussion presented in the responses to reviewers (that a $k$-NN graph is constructed from point cloud using certain package, then the spectral distance methods are applied to this graph, etc) was missing, and its role was not clear to the readers.
>
> Regarding the ripser library, I agree a detailed explanation of the package is  certainly not necessary. But, since the study heavily depends on this package,  at least a few sentences on what does it take as inputs and arguments (e.g., the maximal feature dimension), how are the resolution scales defined for persistence (like how many levels does it consider), what does it output for persistent homology (is it a list of holes, how do we get m-th persistent features, etc) should be provided. Right now, we just have a single sentence about this. Without these basic details, it is very difficult to understand the methodology used, right?
>
> Same with the main premise of the paper. The two sentences picked by the authors above might allude that the paper does some kind of comparison. However, the paper is not set up as "a comparison study of existing methods for detecting holes in the presence of high-dimensional noise". There are other contradicting sentences which imply differently. For examples,  in the intro, it says "we suggest to use persistent homology with distances based on the spectral decomposition of the kNN graph Laplacian,
> such as the effective resistance ....", which can be interpreted as a new proposal which was not explored before. In the abstract, the term "As a remedy, .." (which is conveniently skipped above) suggests a new solution is proposed.
> I feel some of the comments by the reviewers are rooted in this fact, that this main premise (which is well explained in the general comment/response by the authors) was not really clear in the paper.

---

> > ### Author Response · Authors · 2023-11-22
> >
> > Thank you for getting back to us again and pushing for a clearer exposition. We will take your comments under consideration and aim for further clarifications in future revisions.
> >
> > We would only like to comment on one aspect, to prevent misunderstandings:
> >
> > > in the intro, it says "we suggest to use persistent homology with distances based on the spectral decomposition of the kNN graph Laplacian, such as the effective resistance ....", which can be interpreted as a new proposal which was not explored before.
> >
> > But indeed, it _is_ a new proposal, and it was not explored before!
> >
> > As a result of our benchmark, we found that spectral methods help persistent homology to detect holes despite high-dimensional noise. While these distances themselves are not new, as we do acknowledge, the combination of *corrected* effective resistance with persistent homology is novel and is one of our key contributions. We clearly state this in the relevant work section.

---

### Official Review · Reviewer_TJmS · 2023-10-31

**Soundness:** 2 fair
**Presentation:** 3 good
**Contribution:** 2 fair
**Rating:** 6
**Confidence:** 3

**Summary:**

This paper aims to address the challenge that the persistent homology (PH) faces for noisy high-dimensional data, by replacing the standard Euclidean or other common distances with the diffusion distance and effective resistance.

**Strengths:**

(S1) The problem is relevant.
(S2) The paper is well written, resulting in a nice and enjoyable read.
(S3) Experiments include a number of synthetic and real-world data.

**Weaknesses:**

(W1) Some (main) statements might not be correct or precise enough, placing some doubts on the overall paper, see the questions below. I will raise my score if these issues are resolved.

(W2) The results are not very convincing: for the synthetic data, tSNE and UMAP seem better or equally good as the proposed distances but are for some reason shown only for 1 out of 7 data sets, and for the real-world data it is not clear how the ground truth is established.

(W3) The contribution might not be strong enough for this venue (e.g., PH on diffusion distance or effective resistance is not novel).

**Questions:**

(Q1) Are you trying to address the issue of Gaussian noise or outliers, or both? Be precise and consistent.

(Q2) The main issue with Gaussian noise for high-dimensional data is that the small noise adds up over the many coordinates, for Euclidean l_2 distance. A natural adjustment would be to rather consider l_infty, could you include these experiments (at least for the noisy ring that you consider the most), or at least discuss why this would not be a suitable approach?

(Q3) What do you actually mean with “ring”, the main example used throughout the paper? If this the circle (Figure S1), I would suggest to rather use that terminology. If this is an annulus, then Figure S1 should be adjusted. If by noisy ring you mean a circle with Gaussian noise, then it is probably clearer to use the latter formulation. On a related note, I think it would be good to include an additional figure that illustrates the different levels of noise (e.g. sigma in {0, 0.05, 0.10, 0.15, 0.20, 0.25, 0.30, 0.35}) you consider in the experiments, on (2D MDS embedding of) an example shape (like the circle). This can help to get an idea of the level of noise that the different approaches can handle.

(Q4) In Related work, you write that the previous approaches in the literature amount to replacing the Euclidean distance with a different distance matrix. However, the DTM filtration does not simply do this, it rather considers a weighted Vietoris-Rips filtration with nonzero filtration function values on the vertices too. These values correspond to the average distance from a number of nearest neighbors, so that outliers have a large filtration function value and appear only later in the filtration, i.e., they are smoothed out in the process. This does not influence your results, since you only consider 1- and 2-dimensional PH, and the edges that could create loops and voids appear only after the incident vertices would appear. However, this would make a huge difference for 0-dimensional PH, since the outliers would result in persistent connected components with Vietoris-Rips filtration, the issue which is avoided with DTM (which is not clear with your current description). This needs to be made more precise. Revise if this is the case for other filtrations too.

(Q5) How do you define the adjacency matrix A in your experiments?

(Q6) Has the corrected version of the effective resistance also already been introduced in the literature (if so, provide a reference), or is this your contribution?

(Q7) Why do you not show the results for tSNE and UMAP in Figure 5? These seem to perform extremely well on the noisy ring (Figure 4), and definitely better than Fermat or DTM that you do include. Overall, you show results for different subsets of the 12 distances across different figures.

(Q7) When discussing the other approaches in the literature (such as Fermat or DTM), could you make it more explicit that these have not been introduced to tackle the particular issue that you aim to address (noise in high dimensions)? (For example, the idea behind DTM is to smooth out the outliers out, but you do not seem to consider these in your experiments?) Otherwise, it seems you are overstating your contribution, as it appears that you outperform the other approaches that were developed to tackle the same challenge. You even imply this by referring to the other methods as “competitors”.

(Q8) You group the different distances into density-based, graph-based (distances computed on kNN graph), embedding-based and spectral. However, are the DTM, Fermat and Core also not computed on the kNN graph (whereas you consider these to be density-based)? In the Discussion, you also explicitly write that spectral methods are based on the kNN graph, so I guess you need to rethink the naming and descriptions of the different groups?

(Q9) The two proposed distances, effective resistance and in particular diffusion, fail terribly in detecting both the 2 loops and the 1 void for torus, even in the case of no noise (Figure 5), but you almost completely ignore this?

(Q10) Figure 5 shows a large variance for the diffusion distance and effective resistance, in particular for eyeglasses and torus. This should be at least briefly discussed, and it would be nice to include an illustrative figure with a few different random walks between two interesting points (e.g. on eyeglasses); ideally, other distances could be visualized too.The number of different random walks also grows with the underlying dimension, since there is many more directions one could take to reach from one point to another? Can you also comment on this, because it makes one wonder why would such distances be reasonable/even suggested for very high dimensional spaces?

(Q11) Where is the variance for Euclidean distance coming from in the left plot of Figure 6?

(Q12) Why do you consider the 2D embedding space for the embedding-based distances, if you also look into 2-dimensional PH?

(Q13) Why is the closed-form formula for effective resistance useful (in your experiments or work)? Motivate this/explain the relevance.

(Q14) What are the dimensions of the real-world RNA-sequencing data? This should be explicitly stated, since high dimensionality is precisely the main focus of your work. This is only mentioned for 1 out of 6 data sets and only in the appendix, but this information should be in the main text.

(Q15) “… DTM produced only rough approximations (Figure 8b)” What exactly do you mean here, the 1-dimensional PD wrt DTM in Figure 8 clearly identifies the two loops? What’s more, it seems that the loop score s2 would be the best for DTM, since the second most persistent loop is here the furthest from the third most persistent loop (close to the diagonal)? I do not see a clear connection between Figure 8 and first plot in Figure 9.

(Q16) How do you assess the ground truth (the actual number of loops) in the real-world data (besides Malaria data)?

(Q17) You write “persistent loop(s) was/were likely not correct”, or later, “arguably incorrect loops” but you do not explain this further. In other words, how do you determine if a bar in Figure 9 is hatched, since, as you yourself write “each homology class has many different representative cycles, making interpretation difficult”?

(Q18) Definition of the DTM function in Appendix B is weird, can you provide a reference? In the paper by Anai et al, it seems that only the case of your p=2 is considered? Are the nearest neighbors x_i1, x_i2, …, x_ik ordered according to increasing distance from x_i? If so, please specify. How does it make sense to define dtm_i = ||x_i – x_ik || (when your p=infty)? Strangely enough, all your main experimental results consider p=infty. On a related note, it is not clear to me why the DTM performs worse than the Euclidean distance in Figure S4? This makes me question how you chose the parameter values for which to report the results, and whether you particularly selected the parameters where the other approaches perform poorly.

(Q19) “We omitted DTM as all settings got filtered on all datasets.” What does this mean?

(Q20) Interpret all the figures in Appendix G, what do we learn from them? This is currently only done for Figure S3, in its caption.

(Q21) Why is tSNE performing so poorly in Figure S5, even when there is no noise?

(Q22) What about stability?

(Q23) Multiparameter persistence is often suggested to remedy noise, by considering a bifiltration with respect to both distance to the point cloud and density estimates. How do you expect this approach to work in high dimensions?


Minor remarks:

-	Mention the homological dimension in the captions of all figures that include persistence diagrams (i.e., stress “1-dimensional persistence diagram”).

-	“…. distances due to noise dominate the distances to the ring structure” Is this really true, we can still see the ring in Figure 3d?

-	Describe the hitting time H_ij more precisely. Is the the sum of edge lengths, or the number of edges?

-	I assume that the matrix I_d is  matrix of ones (every entry is equal to 1), but this should be made explicit, as this notation is common for the identity matrix (with the non-diagonal entries equal to 0).

-	For consistency and clarity, replace “neg. control” with “0 loops” in Figure 5?

-	When you mention t in Section 7, remind the reader what this t represents. Note also that you use t to denote both the filtration scale (could maybe be replaced with r), and for the number of random walk steps.

-	… D = 2, but D is a matrix?

-	Provide a reference for the computational complexity for PH. Should it include delta+1, or delta+2?

-	Be consistent between capital case vs. lower case for the paper titles in the References.

-	The notation in Appendix C could likely be improved: the distances are functions over the vertices rather than of its parameters, i.e., it would be more common to denote e.g. Fermat and DTM respectively as d^F_p(x, y), D^DTM_k, p, xi(x, y).

-	In Appendix D, for the eyeglasses data set you write that the two line segments are of length 0.53, separated by 0.7 units linking up the two ring segments, but the width of the rectangle in Figure S1 seems larger than its height?

-	For better clarity “, and then added isotropic Gaussian noise samples from …” should probably be the last sentence in this paragraph, since the rest of it discusses the orthogonal embedding?

Typos:

-	naïve -> naive throughout the paper?
-	persistence homology -> persistent homology
-	we mapped each… -> We mapped each

---

> ### Author Response · Authors · 2023-11-17
> **Thank you for the thorough review! (1/n)**
>
> Dear reviewer TJmS,
> many thanks for your very detailed and helpful review! We really appreciate all the concrete suggestions and questions. They prompted many improvements to the manuscript. We will address your questions below.
>
> **(W1) Clarifications:**
> We clarified numerous aspects of the paper, and maintain that our statements are correct. If any questions remain, we are happy to address them as well.
>
> **(W2) Performance of $t$-SNE and UMAP:**
> Please see our general comment on the performance of $t$-SNE and UMAP (as this point came up in several reviews).
>
> **(Q7a) Exclusion of embedding methods from Figure 5:**
> We excluded the $t$-SNE and UMAP results from Figure 5 partly for the above reasons and partly because this already busy figure would have become messy otherwise.
> We ran a large number of experiments on several datasets. Therefore, we deliberately chose different methods and datasets for the figures in the main text to illustrate the key take-aways of our analysis in a concise way. Note that we show the results of all methods on all dataset, inlcuding $t$-SNE and UMAP in the added supplementary figures S10-S21.
>
> **(Q21) $t$-SNE performance with low perplexity:**
> The reason for the dip of the $t$-SNE embedding method for low noise and low perplexity is a curvy embedding of the circle with various bottlenecks that give rise to additional features of high persistence. We included a new Figure S9 explaining this in more detail.
>
> **(W3) Novelty:**
> Please, see statement on novelty to all reviewers.
>
> **(Q1) Focus of the paper:**
> Our work addresses high-dimensional Gaussian noise, however we added an experiment with outliers in the revised version, following reviewer BdHd's concern (Appendix G). As stated at the end of Section 4 and the beginning of Section 9, moving from low to high ambient dimension, outliers cease to be an issue for persistent homology, instead Gaussian noise becomes problematic. For this reason, we focus on Gaussian noise. As can be seen in the new Fig. S6, adding up to 10% outliers impacts all methods only minimally in high ambient dimension. While not the focus of our work, please note that effective resistance also handles outliers better than Euclidean distance in low ambient dimesion (Figure S6).
>
>
> **(Q7b) Adequate presentation of DTM and Fermat:**
> You are right that DTM and Fermat distances were introduced to handle outliers, as we write in the related work section and Section 7. We have added a sentence in Section 7 stressing that we apply them to a different type of noise in our paper.
>
> **(Q2) Using $\ell_\infty$ to deal with noise:**
> Thank you for this suggestion. We ran experiments on our toy datasets for $\ell_p$ distances with $p=3,5,7, \infty$. The results are close to or worse than for the Euclidean distance (Figure S21). We believe that the reason for this is that for any $p\neq 2$ the $\ell_p$ distance favors directions and is not isotropic anymore. However, we place our toy datasets in a random orientation in ambient $\mathbb R^{50}$, so that the directions in which the dataset varies meaningfully are not aligned with the coordinate dimensions.
>
> **(Q3) Term "ring":**
> Thank you for spotting our imprecision. When speaking of "ring" we always meant a circle (plus Gaussian noise). We have changed all uses of "ring" to a more precise phrasing, as suggested.
>
> Moreover, we have included Figure S4 that shows the circle with different levels of Gaussian noise in 2D as well as 2D MDS embeddings of the circle with different levels of Gaussian noise in 50D.
>
> **(Q4) Node weights in DTM:**
> We deliberately omitted this detail since, as you acknowledge, it does not matter for our experiments and omitting it simiplified the exposition. In the interest of accurarcy, we have added this aspect to the detailed description of DTM in Appendix C. We tweaked the relevant sentence in the related work to "The main idea of most of these suggestions is to replace the Euclidean distance with a different distance matrix, before running persistent homology. "

---

> > ### Author Response · Authors · 2023-11-17
> > **Thank you for the thorough review! (2/n)**
> >
> > **(Q15 & Q19) Performance of DTM:**
> > "Rough approximation" meant the representatives, which only consists of a handful of edges for both loops. Such a represenative can be misleading to a practitioner. Nevertheless, we cautioned against the overinterpretation of representatives in the Discussion.
> >
> > The persistence diagram for DTM in Figure 8b does indeed look good. Its detection score is about $0.88$ and thus lower than those of effective resistance ($0.93$) and diffusion ($0.99$). DTM's bar is not visible in Figure 9 because of the heuristic we called "filtering" (paragraph "Performance metric" in Section 7). We omit an entire persistence diagram (treat it as if no features were found) if the death over birth ratio for all features is below a threshold of $1.25$. As explained in Section 7, the reason for this heuristic is that some methods sometimes produce diagrams with very few features, sometimes just a single one. Such a diagram could indicate a single "real" feature and no noise features, or just a single noise feature. This makes such a diagram difficult to interpret and our detection score to have large variance. So we decided to threshold all features by their death/birth ratio.
> >
> > We omit this heuristic in Figure S12 to illustrate its effect. For instance, in the high noise regime the curve for $t$-SNE with $\rho=333$ jumps up and down, that for core distance with $k=100$ has very large standard deviation and that of the Euclidean distance increases (while detecting an incorrect loop as most persistent one). All of these problems get resolved by the heuristic (compare Figures S10 and S12).
> >
> > It was common to the problematic cases that the few features that we found lay very close to the diagonal in the sense that their death-over-birth time ration was small, leading us to our heuristic.
> >
> > DTM also often produced persistence diagrams with very few points, frequently corresponding only to noise features. However, the death-over-birth ratio for points in persistence diagrams of DTM were also generally very low, so that it can get strongly affected by our heuristic. In particular, on all single-cell datases no persistence diagram produced with any hyperparmeter configuration of DTM passed the thresholding, such that there is no entry for DTM in Figure 9. This is also what is meant by the sentence "We omitted DTM as all settings got filtered on all datasets."
> >
> > We realize that in the context of persistent homology with its use of a filtered simplicial complex, it is not ideal to call our heuristic "filtering" as well. We have changed its name to "thresholding" throughout the paper in the revision.
> >
> >
> > **(Q18) Definition of DTM:**
> > A more general definition of distance-to-measure using arbitrary $p\geq 1$ was given in Chazal, Frédéric, and Bertrand Michel. "An Introduction to Topological Data Analysis: Fundamental and Practical Aspects for Data Scientists." Frontiers in artificial intelligence 4 (2021): 108. Thanks for pointing out that this reference was missing. We added it to Appendix C.
> >
> > We explored the DTM-based Vietoris-Rips both in Anai et al. 2020's original setting ($p=2$) as well as in the additional $p=\infty$ setting, in order to strengthen the baseline for our spectral methods. In the figures in the main text, we always reported the *best* hyperparameter setting for every method (measured by the area under the detection curve for toy experiments and by the detection score for single-cell dataset), as explained in Appendix E. In Appendix E, we also give details on the hyperparameters used for each figure.
> >
> > On some datasets (e.g. Linked circles) the $p=\infty$ setting performed best, while on others (e.g. Circle) the $p=2$ setting performed best. In particular, most of our main figures show the $p=2$ case, as explictly stated in Figures 5 and 7. Note that for DTM we actually explored more than four times more hyperparameter combinations than for any other method.
> >
> > The reason DTM performs worse than Euclidean in Figure S10 is that this figure employs our filtering (aka thresholding, see above). Note that we provide an additional Figure S12 where we do not use this heuristic and indeed DTM outperforms the Euclidean distance, but only slightly.
> >
> > The nearest neighbors were indeed meant to be ordered and we have added this to the text. Choosing the distance to the $k$-th nearest neighbor for $p=\infty$ is simiply the limiting case of taking $p\to \infty$ (similar to how $\ell_p \to \ell_\infty$ for $p \to \infty$).

---

> > > ### Author Response · Authors · 2023-11-17
> > > **Thank you for the thorough review! (3/n)**
> > >
> > > **(Q5) Definition of adjacency matrix:**
> > > We define the adjacency matrix $A$ of an undirected, unweighted graph as the symmetric matrix with entry $a_{ij} =1$ if edge $ij$ is part of the graph and $a_{ij} = 0$ otherwise. If the graph is weighted, then the non-zero entries of $A$ are set to the corresponding edge weights. We added a definition to Section 5 and Appendix A.
> > >
> > > **(Q6) Contribution to the corrected effective resistance:**
> > > The corrected version has been proposed by Ulrike von Luxburg, Agnes Radl, and Matthias Hein. Getting lost in space: Large sample analysis of the resistance distance. Advances in Neural Information Processing Systems, 23, 2010, as we mention in the related work, and Sections 5, 6. It was also known that it is possible to realize the distance as squared Euclidean distance, yet the proof was not constructive (Prop. 4 in the above reference). As a result, no explicit formula realizing this distance as squared Euclidean distance was known. Our contribution is in working out this explicit formula. We wrote in Section 6 that "Here we show that the corrected effective resistance (von Luxburg et al., 2010a) can also be written in this form [explicit square of euclidean distance] (see the proof in Appendix A):" which clearly indicates what was known and what we contributed.
> > >
> > > **(Q13) Usefulness of the explicit formula for corrected effective resistance:**
> > > The explicit formula for the corrected effective resistance is useful on a theoretical level to compare and better understand the approaches taken by different spectral methods in Section 6. The explicit formula shows how corrected effective resistance decays the eigenvectors of the graph Laplacian differently than diffusion distance and the uncorrected version. For instance, it decays  eigenvectors of high eigenvalue more aggressively than the uncorrected version, but less aggressively than diffusion distances.
> > >
> > > **(Q8) Names of the groups of distances:**
> > > You are right, all methods save for Euclidean and Fermat distances rely in some way on the $k$NN graph. We hoped to structure the different explored methods by dividing them into named groups, but this rather seems to create confusion. We have omitted the subdivision into groups and their names from Section 7 and Figure 5.
> > >
> > > **(Q9) Performance on the torus:**
> > > While we maintain that effective resistance is on par with the Euclidean distance on the torus, this is indeed an interesting observation! We investigated this further and found that the sample size was too low for the torus. Increasing the number of points from $n=1000$ to $n=5000$ improved the performance of all methods for loop detection, but in particular the performance of the spectral methods which clearly outperform the others in this setting, see the new Figure S18, which we now reference in Section 7.1.2. For this high sample size, however, void detection is not feasible for any method. We mentioned a possible way of making non-Euclidean distances profit from higher sample sizes without increased computation as future work in the discussion.

---

> > > > ### Author Response · Authors · 2023-11-17
> > > > **Thank you for the thorough review! (4/n)**
> > > >
> > > > **(Q10) Uncertainty of spectral methods:**
> > > > The uncertainty of effective resistance and diffusion distance on the eyeglasses dataset is because there was a bridge between the straight segments for some random seeds, while not for others. As a result, the second most prominent loop had either very high or very low persistence. We added Figure S8 to illustrate this for the effective resistance. We view the behavior of the diffusion distance for void detection on the torus as a failure case, but believe that a denser sampling would resolve this problem, see our reply to Q9.
> > > > We added Figures S1 and S2 to illustrate the various different distances on the toy circle in $\mathbb R^{50}$ in a similar way as in Figure 3c-f.
> > > >
> > > > We do not think that the number of random walks in high dimension is problematic.
> > > >
> > > > First, the methods relying on random walks perform very well empirically for the high ambient dimensions.
> > > >
> > > > Second, we always perform random walks on the symmetric kNN graph, whose number of edges is upper bounded by $2kn$, independent of the embedding dimension. Of course, its exact structure depends on the noise and the embedding dimension. But according to our analysis in Section 6 the first two eigenvectors of the graph Laplacian for the noisy circle in 50D are much smaller than the others even for high noise levels, such as $\sigma=0.25$ (Figure 4 a). This means the graph is essentially two-dimensional. Based on these eigenvalues, both effective resistance and diffusion distances are able to decay all but the first two eigenvectors for appropriate hyperparameter values (Figure 4 c-e).
> > > >
> > > > Third, on all but the simplest graphs the number of random walks that effective resistance considers is always infinite, as the random walker can alternate back-and-forth between two nodes arbitrarily often. Note that to compute the effective resistance, we do not need to sample these random walks explicitly, but only need the pseudoinverse of the graph Laplacian.
> > > >
> > > >
> > > > **(Q11) Uncertainty of Euclidean distance in Figure 7a:**
> > > > The uncertainty for the Euclidean distance in Figure 7a stemmed from the randomness in the Gaussian noise that we add to the circle. We always show means and standard deviations over three random seeds, see Section 7.1.1.
> > > >
> > > > **(Q14) Dimensionality of the single-cell datasets:**
> > > > We gave the dimensionalities of all single-cell datasets in Appendix D.2 and mentioned their range as $10-5126$ in Section 7.2. To make this even more explicit we added the dimensionalities to Figure 9.
> > > >
> > > > **(Q16) Ground truth for the single-cell datasets:**
> > > > We selected single-cell datasets for which the biology of the cells tells us what topological features to expect, so that ground truth is known. For the malaria dataset there are two cycles: the parasite reproduction cycle in red blood cells and the parasite transmission cycle between humans and mosquitos as described in Section 7.2. The other five datasets are created so that the gene expression variation comes from the cell division cycle (mytosis), that is, the process of a cell dividing into two copies of itself. Therefore, we expect one loop in these datasets.
> > > >
> > > > **(Q17) Evaluation of representatives:**
> > > > We manually compared the representatives of the most persistent loop(s) to meta data provided by the papers from which we obtained the datasets. This meta data included visualizations, cell type annotations, and estimations for the progression along the cell cycle, like in the colored visualization in Figure S5. While we tried to be fair and charitable, this was admittedly a subjective procedure, which is why we flagged it so heavily ("manual", "arguably", "making interpretation difficult").
> > > >
> > > > Nevertheless, we think that it is important to point out that we encountered several cases where a method might predict the correct number of loops but the representative returned by the ripser package is a poor, if at all correct, approximation of the ground truth cycle.
> > > >
> > > > **(Q20) Interpretation of supplementary figures:**
> > > > We made an effort to interpret the supplementary figures in more detail. However, given the large number of experiments we cannot comment on every finding. We included the figures S10-S21 reporting results for many hyperparameter values of all methods as backdrop for the interested reader as well as to be transparent about our full results.
> > > >
> > > > **(Q22) Stability:**
> > > > We ran all our experiments for three random seeds and report means and standard deviations, as stated in Section 7. As can be seen in Fig. 5, effective resistance and diffusion usually outperform the other methods in such a way that standard deviations do not overlap. High uncertainty typically happens mostly in the high-noise regime, as is to be expected.

---

> ### Author Response · Authors · 2023-11-17
> **Thank you for the thorough review! (5/n)**
>
> **(Q23) Multiparameter persistence:**
> Multiparameter persistence is an interesting method and might offer benefits in the high-dimensional setting. It has been employed to spatial transcriptomics data in  Benjamin, Katherine, et al. "Multiscale topology classifies and quantifies cell types in subcellular spatial transcriptomics." arXiv preprint arXiv:2212.06505 (2022).
>
> However, its output is not as readily interpretable as the persistence diagrams in single parameter persistent homology, which is why we limited out study to this more established method.
>
>
> **Minor 1. Miscellaneous:**
> We added the feature dimension to all persistence diagram, changed the filtration parameter to $\tau$, changed the notation in Appendix C, replaced "(neg control)" by "(0 loops)", and inserted a line break after “, and then added isotropic Gaussian noise samples from …”. Thank you for these suggestions!
>
> **Minor 2. MDS of highly noised circle:**
> It is true that one can still see a hole in the middle of Figure 3d. But this is a 2D MDS and cannot capture the exact shape of the data in $\mathbb R^{50}$. Indeed, we see in Figure 3c that the feature for the correct loop disappears into the cloud of noise features (green) and in Figure 1a,b we see that the most persistent loop actually does not wrap around the noisy circle.
>
> **Minor 3. Definition of hitting times:**
> The hitting time $H_{ij}$ is the expected number of edges that a random walker starting at node $i$ traverses before reaching node $j$ for the first time, as stated in Definition 2. We changed the ambiguous term "average time" in the main text to "average number of edges".
>
> **Minor 4. Notation for identity matrix:**
> By $\mathbb I_d$ we always denote the $d\times d$ identity matrix and never the matrix of all ones.
>
> **Minor 5. Overloaded variable $D$:**
> We used $D$ both to denote the embedding dimension of Laplacian Eigenmaps and the degree matrix of the symmetric kNN graph. We changed the former to $\tilde{d}$. Thanks for your keen eye!
>
> **Minor 6. Complexity of persistent homology:**
> We added Myers, Audun D., et al. "Persistent homology of coarse-grained state-space networks." Physical Review E 107.3 (2023): 034303 as reference for the computational complexity of persistent homology. The term $d+1$ is correct: The complexity of the reduction algorithm is $\mathcal{O}(N^3)$, where $N$ is the number of simplices in the simplicial complex. A Vietoris-Rips complex can contain at most $\binom{n}{\delta}$ simplices of dimension $\delta$. To compute $\delta$-dimensional homology one needs the simplices of dimension $\delta-1, \delta, \delta+1$. Finally,
> $$
> \sum_{i=1}^{\delta+1} \binom{n}{i} \leq \sum_{i=1}^{\delta+1} n^i \in \mathcal{O}(n^{\delta+1}).
> $$
>
> **Minor 7. Capitalization of references:**
> We have originally capitalized the titles of references as in the original publications, but we are going to change it to a unified format in the next days.
>
> **Minor 8. Dimensions of the eyeglasses dataset:**
> You are right about the eyeglasses dataset. The length of the straight segments is actually $1.06$. We corrected the text.

---

> > ### Comment · Reviewer_TJmS · 2023-11-21
> >
> > I appreciate and thank the authors for the very detailed responses. If possible and allowed by the Area Chair, could you please update a revised version with all the improvements you made clearly highlighted? Unfortunately, I did not have the time to go through all the new results in detail, but let me at least provide a summary of main comments for now:
> >
> > -	Describe your contribution and novelty more clearly in the paper, as you do in the general comment here.
> > -	The main idea behind introducing Fermat is not (only) to tackle outliers, please correct this (in Section 7, and elsewhere if needed). In the original paper, you can see that the authors mainly want to incorporate the information about the density, to e.g. detect a single 1-dimensional loop in the eyeglasses data even when no noise is present (rather than 2 loops with the Euclidean distances).
> > -	Some of the figures in the revised version, such as Figure 3, S2, S4, S5 are for me not displayed correctly?
> > -	You mention a number of times that t-SNE and UMAP are primarily used for 2D embeddings, can you provide a reference? I don’t see why 2D holes could not be assessed from the 3D embedding, have you tried this? On a related note, I don’t think that the comment ‘we wanted to offer alternatives to t-SNE and UMAP, to separate "visualization" from "topological analysis"’ makes a lot of sense, since t-SNE and UMAP are not used for visualization in your paper, but precisely as a part of the TDA, i.e., PH pipeline (to calculate the input distance matrix).
> > -	Adjacency matrix: What exactly do you mean with “if edge ij is a part of the graph”? If i and j are point cloud points, the weight of the edge between them is defined as their distance, i.e., we could consider a fully connected graph?
> > -	Identity matrix: If I_d is the identity matrix, that means that you add Gaussian noise only in one coordinate for each point cloud, why?
> > -	Unfortunately, I still did not have the time to carefully go through your detailed explanation about the results for the DTM filtration, and the related “thresholding” technique that you use, I hope to be able to do this in the following days.
> > -	When I asked about the stability, I was referring to the stability theorems for PH: How is the upper bound for d(PD(X), PD(X’)), where X’ is the point cloud X with noise, going to be affected when you consider the two distances you propose?

---

> > > ### Author Response · Authors · 2023-11-21
> > > **Tracked changes**
> > >
> > > Thank you for your response. We are going to reply in more detail tomorrow ASAP, but in the interest of time (given that it's only 1.5 days left until the end of the discussion period), I am replying now to the two technical issues:
> > >
> > > > could you please update a revised version with all the improvements you made clearly highlighted?
> > >
> > > I have now updated the Supplementary Materials zip file to include the diff.pdf file (as generated by latexdiff). Note that one of the reviewers suggested to change the order of sections, so there are several large blocks of text highlighted as changed whereas in reality the text was simply moved around.
> > >
> > > > Some of the figures in the revised version, such as Figure 3, S2, S4, S5 are for me not displayed correctly?
> > >
> > > I double-checked now, and for me all these figures look fine, both when viewed inside the Firefox browser as well as in Ubuntu's standard PDF reader. How do they look for you, and using what software?

---

> > > > ### Author Response · Authors · 2023-11-22
> > > >
> > > > Thank you for getting back to us and summarizing your most pressing concerns!
> > > >
> > > > **Improved description of contributions:**
> > > > We have now improved the description of our contributions as follows:
> > > >
> > > > * In the introduction, we added the sentence "We are the first to systematically study the noise sensitivity of persistent homology in high-dimensionality."
> > > >
> > > > * In the related work section, we clarified that "It [the corrected version of effective resistance] has not been combined with persistent homology."
> > > >
> > > > * In the discussion, we added the sentence "We view it as an advantage that these existing methods can handle the important problem of high-dimensional noise."
> > > >
> > > >
> > > > **Aim of Fermat distances:**
> > > > We now describe the motivation behind Fermat distances by "Fermat distances (Fernandez et al. 2023) aim to exaggerate large over small distances to incorporate the density of the data." That said, Fernandez et al. themselves emphasize the outliers aspect in their abstract: "The use of this intrinsic distance when computing persistent homology presents advantageous properties such as robustness to the presence of outliers in the input data [...]."
> > > >
> > > > **$t$-SNE and UMAP are visualization methods:**
> > > > In the original paper introducing UMAP only visualization experiments in 2D were conducted (McInnes et al. 2018). Similarly, the paper introducing $t$-SNE is even called "Visualizing data using t-SNE" (van der Maaten and Hinton 2008) and also only considers 2D embeddings. Moreover, the popular $t$-SNE package that we use, openTSNE (Policar 2019), only implements one- and two-dimensional visualizations because the method's runtime scales exponentially with the embedding dimension.
> > > >
> > > > We could add some UMAP results in higher embedding dimensions in a future revision, but we want to stress once again that given the recently published (and prominent) criticisms by Chari & Pachter (2023) and Wang et al. (2023) we feel it is important to explore persistent homology methods that do not rely on $t$-SNE or UMAP.
> > > >
> > > >
> > > > **Adjacency matrix:**
> > > >
> > > > We state in multiple places, e.g., in the paragraph above the one you cited from, that we use the $k$NN graph. For this graph the presence of an edge is completely well-defined. Spectral methods are defined not only for this graph, though, which is why we kept the definition of the adjacency matrix general. To avoid any confusion, we changed the sentence defining the adjacency matrix to "For a connected graph $G$ with $n$ nodes, e.g., a symmetric $k$NN graph, let $A$ be its symmetric, $n\times n$ adjacency matrix [...]"
> > > > In particular, we *never* compute geodesics, diffusion distance or effective resistance on the complete graph weighted by the Euclidean distance.
> > > >
> > > > **Identity matrix:**
> > > >
> > > > There is some confusion here. Using the identity matrix (or a matrix proportional to it) as the covariance matrix of a $d$-dimensional normal distribution means that the resulting distribution is isotropic in all dimensions. In particular, it is non-constant in each dimension. We add an isotropic, normally distributed $d$-dimensional noise vector to each of our data points. In other words, we add independent, one-dimensional Gaussian noise to each of the $d$ coordinates of each data point.

---

> > > > > ### Author Response · Authors · 2023-11-22
> > > > > **Comments on stability**
> > > > >
> > > > > **Stability:**
> > > > > Thank you for the clarification! We do not have explicit stability statements of the type you ask for and we believe that it is difficult to capture the quality of, e.g., effective resistance, in such a way.
> > > > >
> > > > > To illustrate, consider the setting of the noiseless circle with $n=1000$ points in $\mathbb R^{50}$ as $X$ and the version with noise of standard deviation $\sigma=0.25$ added as $X'$. For both $X$ and $X'$ effective resistance finds a single highly persistent loop and the noise features have low persistence compared to it. So persistent homology with effective resistance is stable with respect to the noise level $\sigma=0.25$. However, it is difficult to capture this with the usual distance metric between persistence diagrams. Normally, stability theorems bound the change in the persistence diagram, e.g., in the bottleneck distance. However, the scale of the effective resistance distance changes a lot with the noise. As a result, the death times of the most persistent loops also change by an order of magnitude between the two noise settings, such that the bottleneck distance between the diagrams for effective resistance without noise ($\sigma=0$) and with noise $\sigma=0.25$ is very large. Using the bottleneck distance  (or the Wasserstein distance) fails to capture the structural similarity of the persistence diagrams for effective resistance with $\sigma=0$ and $\sigma=0.25$.
> > > > >
> > > > > On a related note, the standard stability result that bounds the bottleneck distance between two persistence diagrams computed via the Vietoris-Rips complex by the Gromov-Hausdorff distance between the two point clouds becomes mostly vacuous for the Euclidean distance, because the amount of noise aggregated across all dimensions is a lot: For $\sigma=0.25$ the diameter of $X'$ is typically around $4.8$, making the Gromovo-Hausdorff distance between $X$ and $X'$ at least $(4.8-2)/2 = 1.4$, while the most persistent loop in $X$ only has slightly higher persistence $\approx \sqrt{3} \approx1.73$.
> > > > >
> > > > > Nevertheless, we agree that establishing theoretical statements capturing the excellent stability of effective resistance in high-dimensions is an interesting avenue for future research.
> > > > >
> > > > > Do let us know if you have further comments. Your feedback was very helpful and we believe that the paper has much improved. We would very much appreciate if in the end you decide to increase your score, as you wrote initially ("I will raise my score if these issues are resolved").

---

### Author Response · Authors · 2023-11-17
**General comment**

We thank all reviewers cordially for the time and effort invested into reviewing our manuscript. All references in the rebuttals are based on the revised version, which includes the code, as promised.

**Novelty:**

One common criticism was that our paper is not sufficiently novel. However, several reviewers (TJmS, BdHd) appreciated the importance of the problem we tackle, the detection of topology despite high-dimensional noise. We view it as an advantage that this relevant problem can be solved by a combination of existent methods. Furthermore, we believe that papers critically evaluating a set of methods for a relevant problem are invaluable both to the healthy development of the field and to guide practitioners, as acknowledged by reviewer qoEF.

To highlight the main contributions of our paper:

* We are the first to systematically study how different distances affect the persistent homology performances in the presence of high-dimensional noise.
* While effective resistance has been combined with persistent homology before, this performs terribly even for relatively weak high-dimensional noise. It is necessary to employ the corrected version of effective resistance, which has not previously been combined with persistent homology.
* We provide a new, explicit formula for the corrected effective resistance, which is "of independent interest" (reviewer k35W). We use the formula to explain the relationship between effective resistence and diffusion distances.

**Performance of $t$-SNE and UMAP:**

Computing the topology of a high-dimensional dataset on a 2D t-SNE or UMAP plot can indeed be very effective (for H1 homologies), which we acknowledge, e.g., in Sec 7.2 "Persistent homology based on the $t$-SNE and UMAP embeddings could also often identify the correct loop structure and on average worked better than vanilla persistent homology, Fermat distances, and DTM."

However, embedding-based methods can also be problematic and are often criticized in the literature. For example, reviewer TJmS' enquired about the odd behavior of t-SNE with low perplexity in (Q21). Of particular importance is the choice of the embedding dimension. Clearly, the embedding dimension limits the dimension of topological features that can be found. But t-SNE and UMAP are primarily visualization methods, and are mostly used for 2D embeddings, making it impossible to detect any H2 homologies. For Laplacian Eigenmaps, we varied the embedding dimension with mixed results. In contrast, using diffusion distances and effective resistance allows an implicit "soft" choice of the embedding dimension where the importance of higher dimensions decays gracefully and the number of retained dimensions depends on the structure of the dataset (Section 6).

Moreover, t-SNE and UMAP have been severely criticized, especially in computational biology, in the recent years, e.g., by [Chari and Pachter 2023](https://journals.plos.org/ploscompbiol/article?id=10.1371/journal.pcbi.1011288) because they necessarily distort the data. In fact, the work of [Wang et al. 2023](https://www.cell.com/cell-systems/abstract/S2405-4712(23)00209-0) explictly recommends persistent homology instead of UMAP or t-SNE embeddings for single-cell data analysis. In this work, we wanted to offer alternatives to t-SNE and UMAP, to separate "visualization" from "topological analysis".

---

### Meta-Review · Area_Chair_gW2C · 2023-12-12

**Metareview:**

Summary: The article observes that for data with low intrinsic dimension in a high dimensional ambient space persistent homology is sensitive to noise and proposes using spectral distances to overcome this.

Strengths: Referees found the paper considers a relevant problem and presents several interesting experiments including various datasets.

Weaknesses: At the same time there were some reservations about the presentation and the contribution being sufficiently novel or sufficiently strong. Authors provided responses to an extensive list of questions. I commend the authors’ efforts in addressing the concerns brought up in the reviews. At the same time I observe that the number of concerns and edits is substantial and this alone could merit a new round of reviews. At the end of the discussion, two reviewers maintain a reject recommendation and two a borderline recommendation.

Upon inspecting the article, I find that the presented investigations are interesting and yet the scope of the contributions is relatively narrow. In view of a very high bar for acceptance, I must reject the paper. Referees have provided diverse suggestions how the article could be strengthened.

**Justification For Why Not Higher Score:**

Upon inspecting the article, I find that the presented investigations are interesting and yet the scope of the contributions is relatively narrow.

**Justification For Why Not Lower Score:**

NA

---

### Decision · Program_Chairs · 2024-01-16

Reject